# Glucose-regulated and drug-perturbed phosphoproteome reveals molecular mechanisms controlling insulin secretion

Francesca Sacco[1], Sean J. Humphrey[1], Jürgen Cox[1], Marcel Mischnik[2], Anke Schulte[3], Thomas Klabunde[2], Matthias Schäfer[3] & Matthias Mann[1]

Insulin-secreting beta cells play an essential role in maintaining physiological blood glucose levels, and their dysfunction leads to the development of diabetes. To elucidate the signalling events regulating insulin secretion, we applied a recently developed phosphoproteomics workflow. We quantified the time-resolved phosphoproteome of murine pancreatic cells following their exposure to glucose and in combination with small molecule compounds that promote insulin secretion. The quantitative phosphoproteome of 30,000 sites clustered into three main groups in concordance with the modulation of the three key kinases: PKA, PKC and CK2A. A high-resolution time course revealed key novel regulatory sites, revealing the importance of methyltransferase DNMT3A phosphorylation in the glucose response. Remarkably a significant proportion of these novel regulatory sites is significantly down-regulated in diabetic islets. Control of insulin secretion is embedded in an unexpectedly broad and complex range of cellular functions, which are perturbed by drugs in multiple ways.

[1] Department of Proteomics and Signal Transduction, Max Planck Institute of Biochemistry, Martinsried 82152, Germany. [2] Sanofi Aventis Deutschland GmbH, R&D, LGCR, SDI, Bioinformatics, Frankfurt 65926, Germany. [3] Sanofi Aventis Deutschland GmbH, Global Diabetes Division, R&TM, Islet Biology, Frankfurt 65926, Germany. Correspondence and requests for materials should be addressed to M.M. (email: mmann@biochem.mpg.de).

Diabetes is a complex and heterogeneous condition accompanied by a deterioration of glycaemic control. Chronic hyperglycaemia can lead to a myriad of complications, including coronary artery disease and myocardial infarction, stroke, limb amputation, blindness and kidney failure[1].

Beta cell dysfunctional patterns such as delayed insulin secretory rate to a stimulus or loss of pulsatile secretion of insulin represent the major contributor to the initiation and genetic susceptibility to diabetes[2–4]. Therefore identifying new molecular mechanisms contributing to insulin secretion could improve the understanding, treatment and prevention of diabetes.

Growing evidence suggests that protein post-translational modifications, especially phosphorylation, play a crucial role in controlling glucose-mediated insulin secretion[5,6]. In the beta cells of healthy islet, glucose induces insulin secretion rapidly, within a few minutes and maintained for about an hour. This process therefore likely relies heavily on the modulation of protein activities through signal transduction pathways, rather than controlling gene expression. Additionally, by applying mass spectrometry (MS)-based proteomics, here we demonstrate that key kinases are significantly downregulated in NOD (non-obese diabetic mice) diabetic islets, further supporting the importance of phosphorylation-based signalling networks in the glucose-stimulated insulin secretion (GSIS).

However, few studies have investigated changes to the global phosphoproteome occurring after glucose stimulation in beta cells, and their depth (number of quantified sites), was insufficient to capture many of the important regulatory sites[7,8].

Our group recently described a MS-based phosphoproteomics workflow, termed 'EasyPhos', which now enables streamlined and very large-scale phosphoproteome analysis from limited starting material over multiple experimental conditions[9]. Here we apply EasyPhos to comprehensively quantify the response of the phosphoproteome of murine insulin-secreting beta cells exposed to glucose, and in combination with seven different compounds known to act on different pathways affecting insulin secretion. An in-depth proteomic characterization of our chosen cell line model against pancreatic islets validates the experimental system. We subsequently combined our deep compound-dependent phosphoproteomes with time-resolved phosphoproteomic profiling of beta cells stimulated with glucose, revealing phosphorylation sites implicated in insulin secretion control and gene expression regulation. Remarkably a significant proportion of these novel regulatory sites are significantly downregulated in diabetic islets. We discover an unexpected connection to epigenetic control through the DNA methyltransferase DNMT3A which we functionally follow up by interaction proteomics.

## Results

### MS-based proteomics characterization of NOD diabetic islets.
To elucidate the molecular mechanisms governing glucose-stimulated insulin secretion, we performed deep proteomic profiling of murine pancreatic islets extracted from NOD healthy, pre-diabetic and diabetic mice (Supplementary Fig. 1A). We applied the recently developed iST proteomics workflow[10], combined with label-free LC–MS/MS analysis (Fig. 1a), and processed the results in the MaxQuant environment[11,12]. All experiments were performed in at least biological triplicates, revealing high quantitation accuracy and reproducibility, with Pearson correlation coefficients between 0.68 and 0.93 (Supplementary Fig. 1B). Remarkably our proteomic data enable the unsupervised classification of pancreatic islets according to their diabetic stratification (Fig. 1b; Supplementary Fig. 1C). To decipher how the proteome was remodelled in NOD murine islets at different diabetic stages, we focused on the 759 significantly

modulated proteins (analysis of variance (ANOVA), false discovery rate (FDR) < 0.05), and investigated whether these proteins were enriched for particular biological processes or pathways (Fig. 1c). Remarkably we found that proteins involved in vesicle trafficking and secretion were downregulated in pre-diabetic islets. This observation may shed further light at the molecular level on the decreased insulin secretion capability observed during pre-diabetes[13]. Our data reveal that decreased insulin secretion is accompanied by a concomitant reduction in proteins associated with glucose metabolism, while many proteins involved in the regulation of fatty acids and proteins metabolism are significantly upregulated (Fig. 1c). To our knowledge this is the first proteomic data set describing proteome-wide remodelling occurring in mice at different stages of type 1 diabetes (Supplementary Data 1).

We next investigated proteome changes occurring in the NOD islets that may impact the wiring of signalling networks. Of particular relevance was the finding that the levels of key kinases, including AKT2, RAF1, PKA and CaMK2, as well as several phosphatases were significantly modulated in diabetic islets (Fig. 1d). This highlights the importance of phosphorylation signalling networks in controlling the GSIS.

**Validation of the experimental model.** To further characterize the signalling pathways involved in GSIS, we employed Min6 cells, a widely-used insulinoma beta cell line of murine origin that is capable of eliciting a robust insulin secretion response following acute stimulation with glucose[14,15]. Although there are several alternatives with regards to cell lines for studying beta cell function, there is an ongoing discussion as to how faithfully they represent the biological process of interest. We reasoned that similar expression values of proteins involved in a process between two systems would suggest that these processes are well preserved[16]. To quantitatively address this, we performed deep proteomic profiling of both Min6 cells and murine pancreatic islets, which is their *in vivo* cellular context, and of which they constitute 80% of cellar mass[17]. As in our previous experiments, we employed the recently developed iST proteomics workflow[10], combined with label-free LC–MS/MS analysis processed in the MaxQuant environment[11,12]. This strategy enabled the quantification of about 8,500 proteins in Min6 cells, and about 7,800 proteins in whole pancreatic islets (Fig. 2a; Supplementary Fig. 2B), making this the most comprehensive catalogue of islet-cell proteins to our knowledge. Importantly, key physiological processes, such as insulin secretion, glycolysis and oxidative phosphorylation, are amply and equally represented in both the proteome of pancreatic islets and Min6 cells (Fig. 2b; Supplementary Fig. 2B). Employing the recently developed 'proteomic ruler' method[18] we estimated protein copy numbers across the complete islet proteomes, revealing consistent rank orders between the *in vivo* and *in vitro* systems ($R = 0.98$), as well as excellent agreement between the highest and lowest abundant proteins in both cell lines and islets (Fig. 2c). In islets, we were able to measure copy numbers of all the widely used protein markers of beta cells, as well of alpha, PP, delta and epsilon cells. Reassuringly, beta cells markers, such as Ins, Pdx1, Iapp and Mafa1, are largely equally expressed in Min6 cells and in islets (Fig. 2d). Conversely, hormones secreted by non-beta cells, such as glucagon, pancreatic polypeptide, somatostatin and ghrelin, are substantially more abundant in islets than in Min6 cells. Apart from validating the Min6 model at the level of expressed proteins, these results represent a large resource of protein abundance data for murine beta cells and pancreatic islets for use by the community (Supplementary Data 2).

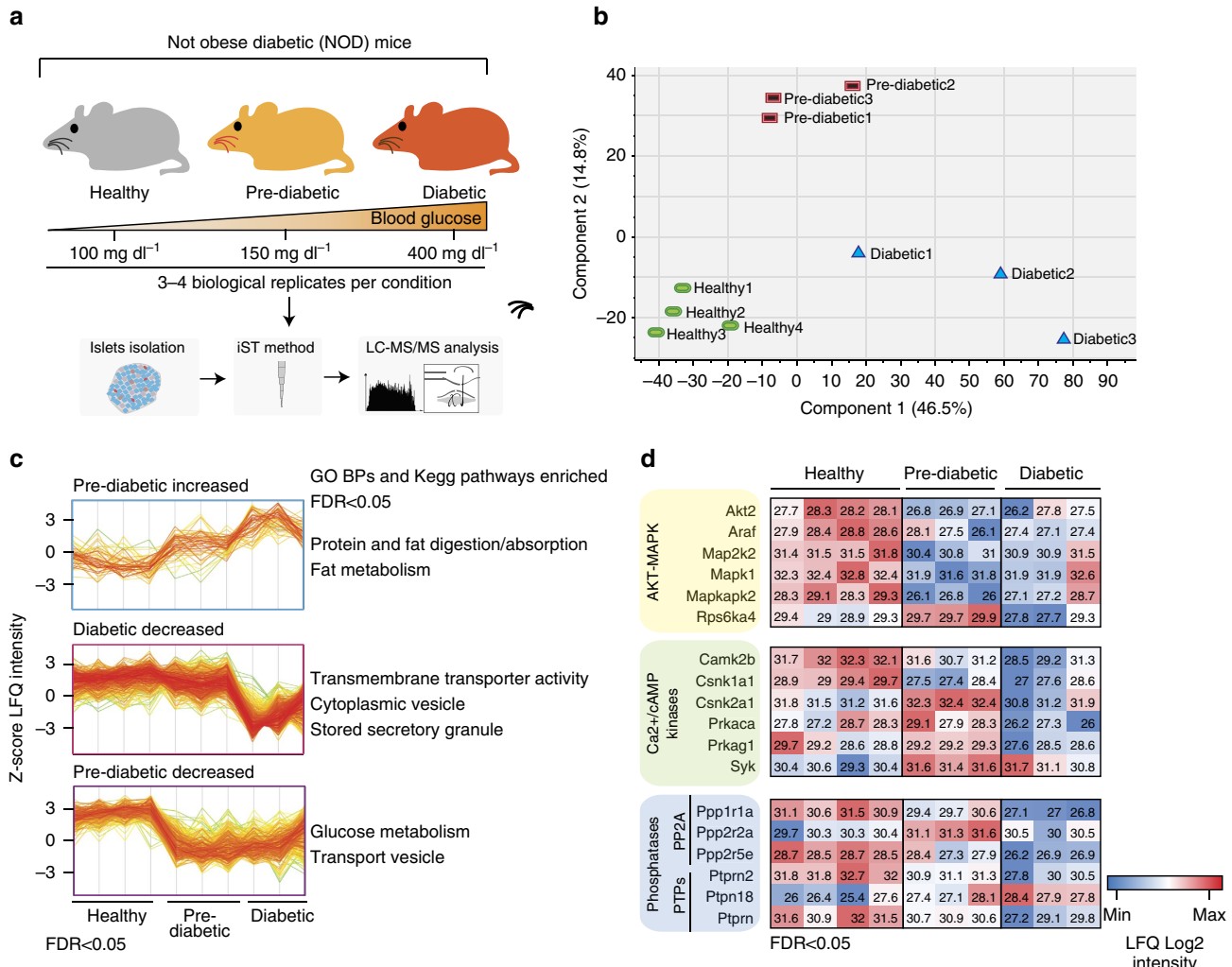

**Figure 1 | MS-based proteomic analysis of NOD islets.** (**a**) Schematic representation of the experimental strategy applied to analyse the proteome of pancreatic islets derived from healthy, pre-diabetic and diabetic NOD mice. (**b**) Principal component analysis (PCA) of NOD islets discriminates the healthy islets from pre-diabetic and diabetic ones. (**c**) GO-Biological processes and pathways significantly enriched in three representative clusters are shown. (**d**) Heat map of kinase and phosphatase protein levels in NOD islets.

**Phosphoproteomics of beta cell glucose and drug stimulation.** To characterize signalling pathways involved in GSIS we next quantified the dynamic response of the phosphoproteome in Min6 cells, by activating or inhibiting the relevant signalling networks in the presence of high or low glucose concentration. To this end, we stimulated the cells by a combination of low or high glucose concentration for 10 min, or high glucose concentration for 30 min. This was combined with one of seven compounds promoting insulin secretion: Glibenclamide, 8-bromo-cAMP, 8-bromo-cGMP, ATP, Carbachol, GLP-1 and the GSK3 inhibitor SB216763, resulting in a total of 25 different signalling states (Fig. 3a). We selected these compounds on the basis of the involvement of their targets in different parts of insulin secretion pathways (Fig. 3b). All experiments were performed in at least biological triplicate. To evaluate the efficacy of each of these conditions, we directly measured insulin secretion after each treatment. All treatments robustly increased insulin secretion, and consistent with published reports[19], secreted insulin reached a peak at 30 min of glucose stimulation (Fig. 3c).

To quantify changes to the global phosphoproteome over these 25 different experimental conditions in great depth and with minimal measurement time, we employed the EasyPhos phosphoproteomic workflow that we recently developed[9]. This resulted in the quantification of more than 35,000 phosphosites located on 5,698 proteins (65% of the total detected proteome). To our knowledge, with the exception of one very large-scale study of mitotically arrested and EGF-stimulated HeLa cells[20], this is the largest quantified phosphoproteome to date. This is made even more remarkable by the fact that previous deep phosphoproteomes have typically relied on extensive fractionation, while the experiments performed here were performed exclusively in single-run mode. Intensity-ranked signal intensities of the 35,058 quantified phosphosites span a wide dynamic range, and phosphorylation sites on very low abundant proteins were also detected (Supplementary Fig. 3A). In total 81% of the phosphoproteome (28,637 of 35,058 sites) was localized with single amino acid resolution (median localization probability 0.999, Supplementary Data 3). In nearly all experimental conditions we quantified >13,000 phosphosites (Supplementary Fig. 3B,C), with a high degree of overlap between them (Supplementary Fig. 3D). Biological replicates measured for each condition demonstrated high quantitation accuracy and reproducibility with Pearson correlation coefficients between 0.75 and 0.90 (Supplementary Fig. 3E,F). Comparison of

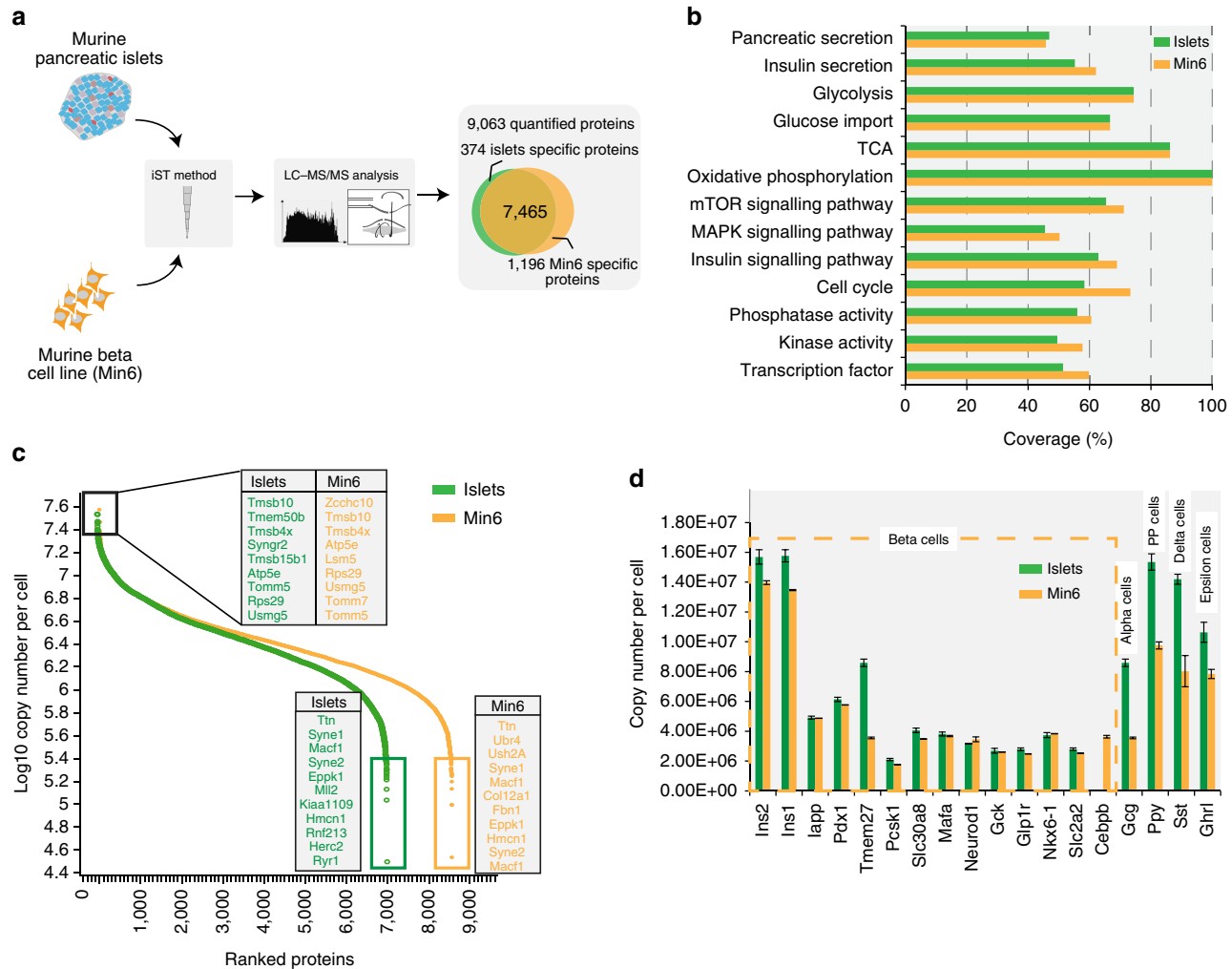

**Figure 2 | The experimental system.** (**a**) Schematic representation of the experimental strategy applied to analyse the proteome of Min6 cells and pancreatic islets. (**b**) Protein coverage in beta cells (orange bars) and islets (green bars). Each bar represents a selected category, for which the number of corresponding protein coding genes in the mouse genome is considered as 100%. (**c**) Ranked protein abundances from highest to the lowest in Min6 cells and islets. (**d**) Copy numbers of protein markers of the different cell populations of islets.

regulatory sites described in the literature suggested a very high degree of coverage of the functional phosphoproteome. To assess this more objectively, we compared the phosphosites quantified in our study with those reported in the PhosphoSitePlus database[21], which revealed 20,209 identical sites—a high proportion considering the depth of our analysis (Supplementary Fig. 3E). Strikingly, about 20% of these phosphosites (4,235 out of these 20,209) were significantly regulated by drug and glucose treatments (ANOVA test FDR < 0.05), as revealed by our MS-based approach (Supplementary Data 3). Remarkably, so far < 3% these sites (329 out of 20,209) have been annotated as 'regulatory sites', revealing that the vast majority of phosphorylation sites quantified here have not been investigated in terms of their functional role or upstream cognate kinase. Of the > 8,000 previously uncharacterized, high-confidence phosphorylation sites, a similar proportion (17%) were regulated by at least one of the drugs (ANOVA test FDR < 0.05), suggesting an even larger scope of functional sites.

To assess whether glucose stimulation triggered the activity of signalling pathways known to play important roles in glucose-mediated insulin secretion in our system, we extracted the phosphorylation sites on the activation loop of kinases of the MAPK and PI3K-AKT pathways. The exposure of beta cells to a high concentration of glucose for 10 min increased the activity of the Mapk1/3 kinases, as well as Foxk1, p70S6K (Rps6kb1) and pS6 (Rps6), considered hallmarks of the RAS-ERK and mTOR pathways, respectively (Fig. 3d(a)). In the same way, we also verified that the drug treatments affected the activity of their targets. As expected, cAMP as well as GLP-1 increased the activity of PKA (Fig. 2d(b,c)); cGMP triggered PKG kinase (Fig. 3d(f)); the GSK3 inhibitor SB216763 decreased the phosphorylation of the GSK3 substrates (Fig. 3d(d)); and Carbachol treatment triggered PKC activity (Fig. 3d(e)). Kinase-substrate motif enrichment analysis of the phosphosites was significantly modulated with respect to the control, independently verifying these findings in each case (Supplementary Fig. 4; t-test, FDR < 0.05). The efficiency of the ATP and Glibenclamide treatments, which do not directly target a kinases, was confirmed by monitoring the phosphorylation of Ptpra, a substrate of CaMK (Fig. 3d(g,h)).

**Unbiased phosphoproteomics discriminates between drugs.** To investigate whether this large-scale multivariate phosphoproteomics data could enable unsupervised classification of small molecule compounds according to their targets, we next

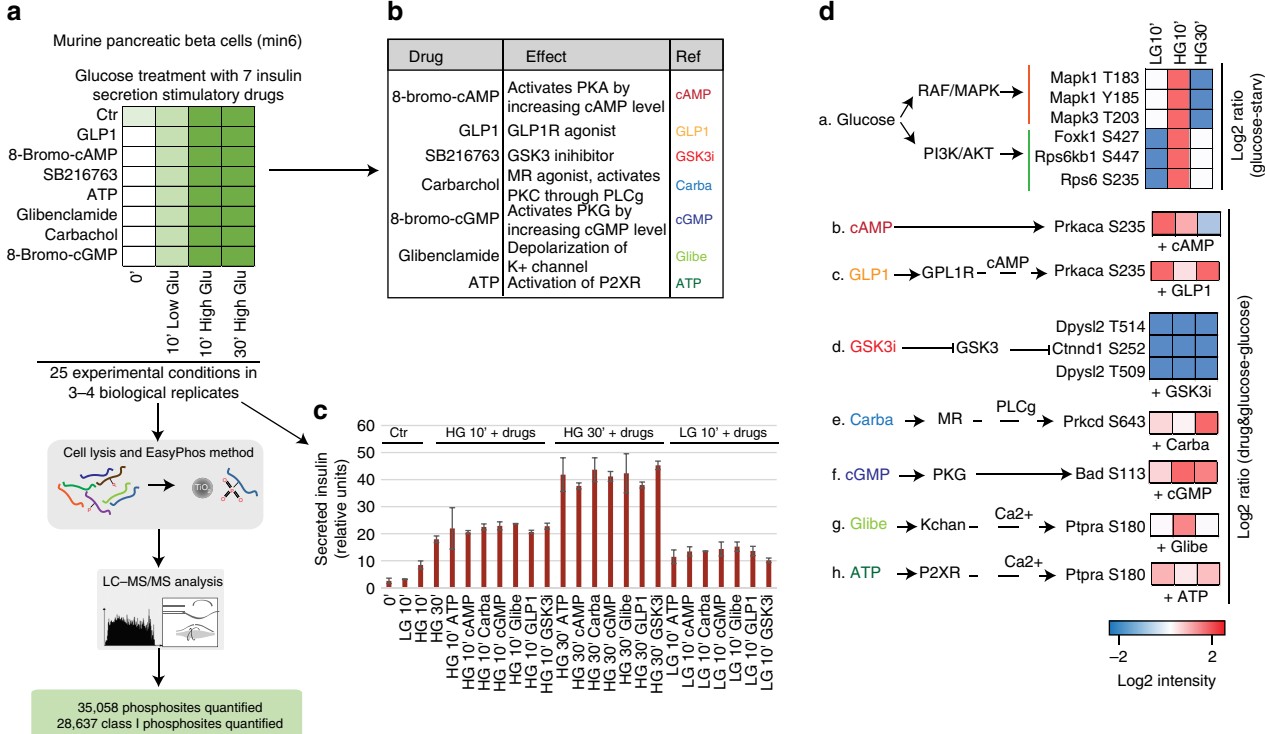

**Figure 3 | Experimental workflow for mapping insulin secretion pathways.** (**a**) Schematic representation of the experimental strategy applied to analyse the changes in phosphoproteome profiles of Min6 beta cells after exposure for 10 or 30 min of low (2.5 mM) or high (16.7 mM) concentration of glucose stimulation in combination with the indicated drugs. (**b**) Drugs used in the study. (**c**) Amount of secreted insulin (a.u.) after drug and glucose stimulation measured with an Elisa assay. Median and s.d. of triplicates are shown as a bar graph. (**d**) Glucose stimulation and drug treatment impair the activity of targeted kinases/pathway.

performed principal component analysis (PCA) of the 22,241 phosphosites quantified in at least 50% of our experimental conditions. PCA segregated the drugs into three main clusters: the first includes drugs increasing the activity of PKA (PKA and GLP1R agonists) together with the GSK3 inhibitor; the second consists of drugs triggering the PKC and PKG kinases (PKC and PKG activators); and the third includes ATP and Glibenclamide, which increase intracellular calcium concentration (Fig. 4a). The fact that a quantitative data set of more than 22,000 phosphosites clearly segregates by mode of action implies that compounds triggering the same kinase/molecule produce similar effects on global phosphorylation. Remarkably, the kinase substrate motifs enriched in the loadings—the phosphosites most responsible for the observed segregation in the PCA—almost always mirrored the expected relationships between drugs and targets, with the additional identification of enrichment of a CK2A substrate motif in connection with the Glibenclamide and ATP compounds (Fig. 4b,c). While PKA and PKC are believed to be essential for GSIS[22], to our knowledge this is the first time that the increased activity of the CK2A kinase is correlated with enhanced insulin secretion. Consistently, the MS-based proteomic profile of NOD diabetic islets also revealed a significant reduction of CK2A levels (Supplementary Data 1; Fig. 1d), highlighting its importance in GSIS regulation

Independent analysis using a correlation matrix (Supplementary Fig. 5A), as well as unsupervised hierarchical clustering of the global phosphoproteome (Fig. 4d) further confirmed the classification of the compounds affecting insulin secretion into these three main groups. Finally, we verified that the activity of kinases, whose substrate motifs were enriched in our analysis, was regulated according to the grouping described above across the many different treatment conditions (Fig. 4e). Thus, the

quantitative phosphoproteomes classify these drugs into three main groups, reflecting their generally similar effects on the pathways represented by three key kinases: PKA, PKC and CK2A (Fig. 4f).

**Mapping signalling pathways triggered by drug treatment.** To decipher how key signalling pathways are modulated by the drugs, we focused on the 6,040 phosphosites significantly modulated in an ANOVA test at a FDR < 0.05 and investigated whether these were significantly enriched for GO-Biological processes (Supplementary Fig. 5B,C) and phosphorylation motifs annotated in the Human Protein Reference Database (HPRD)[23]. This analysis confirmed the kinases that we had previously found to be modulated by the compounds. Interestingly, it also revealed a strong enrichment for substrate motifs of kinases regulating key pro-survival and cell cycle-related pathways, such as MAPK, RAF1, AKT, PIM1 and CDK1/2/3 (Fig. 5a; Supplementary Fig. 5B) (Fisher exact test, FDR < 0.07).

To investigate these observations in greater detail, we applied a recently developed strategy[24] overlaying our phosphoproteomics data onto a literature-derived signalling network that we extracted from the PhosphoSite plus database (Supplementary Fig. 6)[21]. We further filtered this network by maintaining only relationships between proteins that our MS-based proteomic data indicated to be expressed in beta cells (Supplementary Data 2). Using this beta cell-specific signalling network as a scaffold, we overlaid the changes at the phosphoproteome level induced by drug treatment to visualize how the three different classes of drug differentially modulate key signalling networks. (Fig. 5b; Supplementary Fig. 6). While PKA activation (orange nodes) is correlated with increased JNK signalling, drugs triggering the PKC and PKG kinases as well

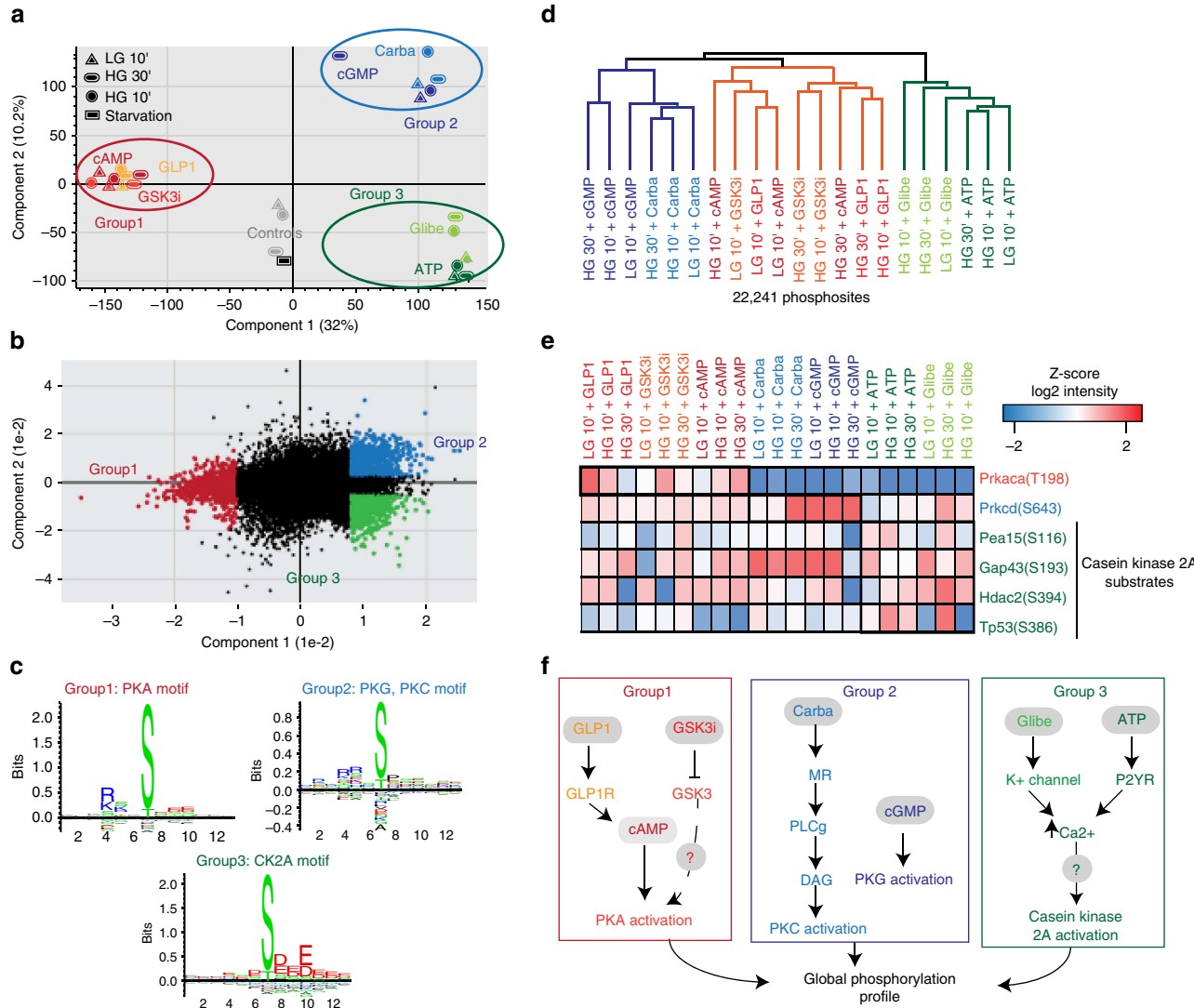

**Figure 4 | Phosphoproteomics data elucidate the relationships between different drugs.** (**a**) Principal component analysis (PCA) of Min6 beta cells treated with different drugs reveals three clusters. Negative controls cluster together and are separated from drug treated cells. (**b**) Loadings of A reveal that the phosphosites responsible for driving the segregation in component 1 and 2 are significantly enriched in (**c**) specific kinase substrate motifs (FDR < 0.05). (**d**) Unsupervised, hierarchical clustering (Pearson correlation distance) of the log2 intensity of 22,241 phosphorylation sites. (**e**) Heat map of phosphorylation levels of the indicated phosphosites upon drug and glucose treatments. (**f**) Schematic representation of the effects of drug treatments in beta cells.

as modulating calcium concentration (blue and green nodes) activate both AKT and MAPK1/2 signalling pathways. While these connections have been described previously, although under different cellular contexts[25–28], our data now provide a global and quantitative phosphoproteomic background to the mechanisms involved.

Interestingly, we found that all the compounds employed trigger pathways that converge upon activation of cell cycle-related kinases, such as Cdk1, Cdk2, Cdk5 and Cdk7. This is consistent with our kinase–substrate motif enrichment analysis (Fig. 5a), which indicated that these compounds may also affect cell cycle progression in addition to increasing insulin secretion. Thus analysis of our deep phosphoproteome provide a molecular framework for the findings that glucose in conjunction with certain diabetic drugs can act as a 'mitogenic hormone' in beta cells and providing a molecular underpinning to the observed connection between insulin secretion and cell proliferation signalling networks[29–31].

**Discovering functional nodes controlling the glucose response.** To extract new functional phosphosites involved in controlling insulin secretion, we started with the 6,040 ANOVA significantly modulated sites (Supplementary Data 4), and selected only those peptides whose phosphorylation was increased after glucose stimulation, and was also further enhanced by treatment with all of the compounds tested (Fig. 6a). This strategy revealed 64 phosphosites on 63 proteins to be hyperphosphorylated by drug treatments. Remarkably, many of the identified phospho-proteins are involved in processes expected to be triggered by glucose stimulation, such as vesicle trafficking, the regulation of calcium channels and the release of insulin granules.

Next, we decided to investigate the temporal regulation of these sites and of the global beta cell phosphoproteome in a glucose-dependent time course in insulin-secreting cells. We again applied the EasyPhos pipeline with increased time resolution of glucose administration, with eight time points spanning from 2.5 min to 1 h (Fig. 6b). Consistent with our previous

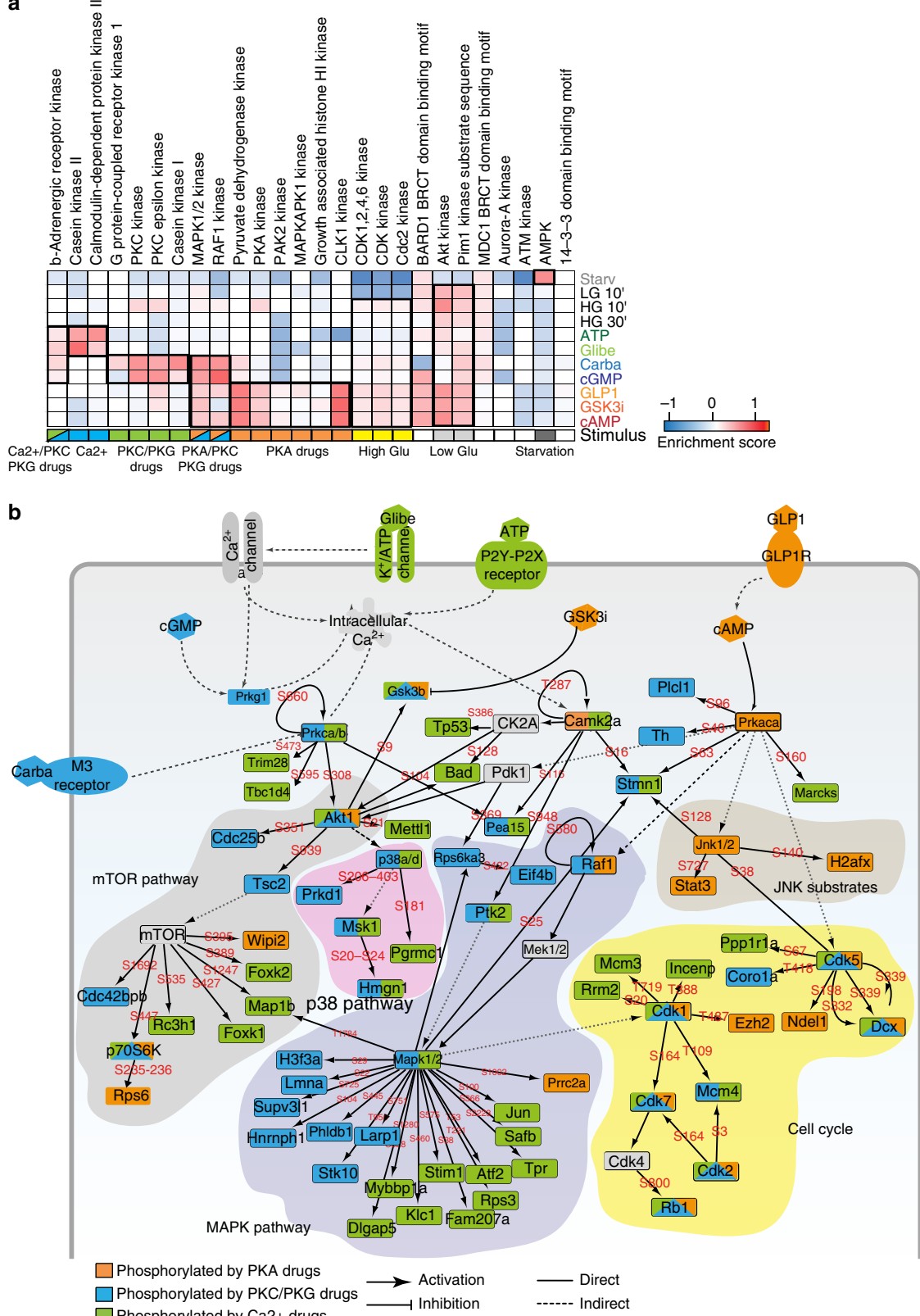

**Figure 5 | Global pathways regulation in response to drug treatments.** (**a**) Heat map of kinase substrate motif enrichment (FDR < 0.02) in glucose and drug-induced phosphoproteomes. (**b**) Phosphoproteomics data were mapped onto a global naive network of signalling information and then filtered (Supplementary Material). Nodes were colour coded according to the class of compounds responsible of their phosphorylation.

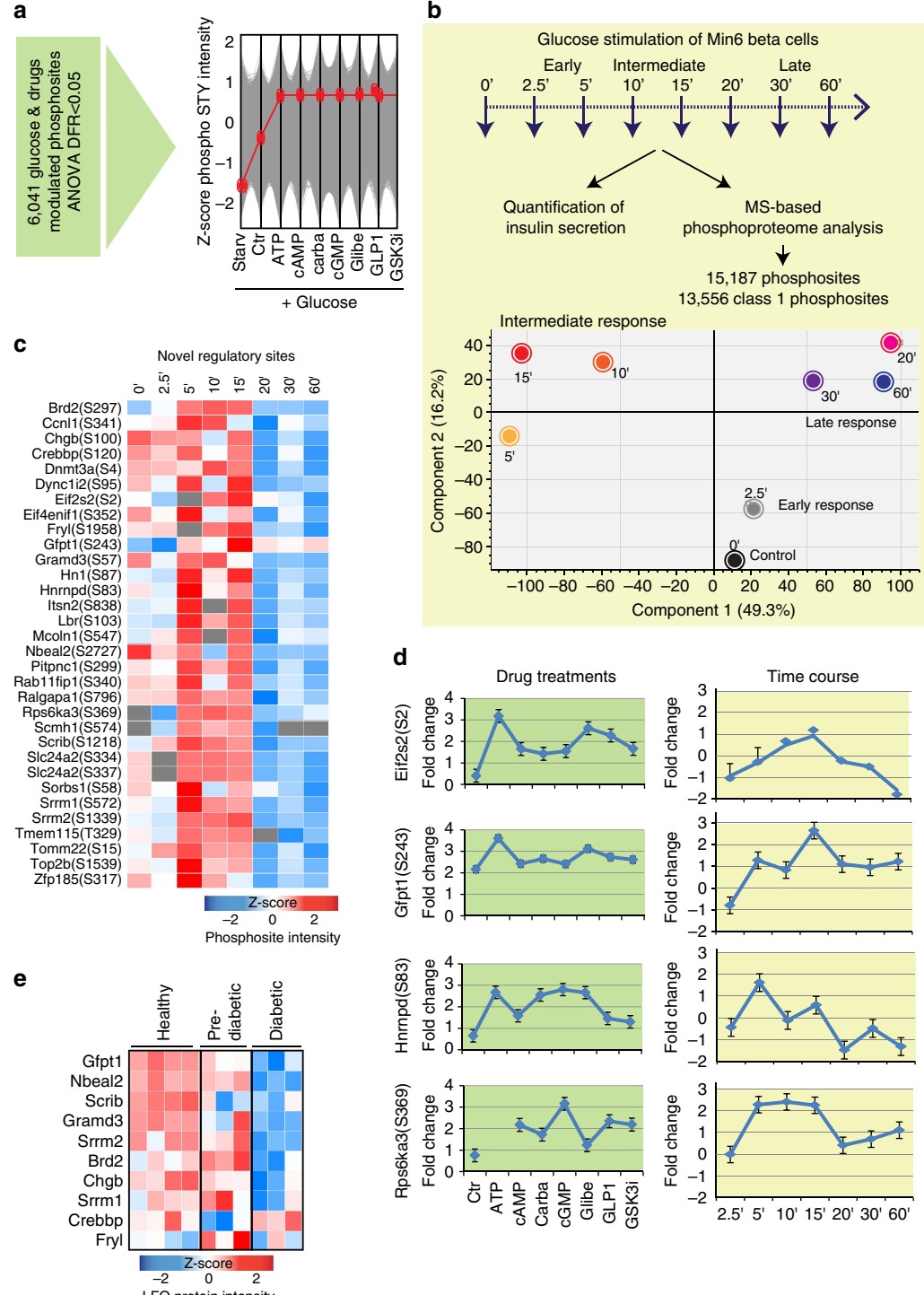

**Figure 6 | Identification of new functional sites controlling insulin secretion.** (**a**) Phosphorylation profile of the potential regulatory phosphosites. (**b**) Experimental strategy applied to profile the phosphoproteome in a glucose time course experiment. Lower part: PCA of Min6 beta cells treated for the indicated time points with glucose. (**c**) Heat map of the Log2 LFQ intensities of the potential regulatory phosphosites significantly modulated in the time course experiment. (**d**) For the four known regulatory sites, the median fold-changes and s.e.m. compared with starvation control cells are plotted. (**e**) Heat map of the Log2 LFQ intensities of 10 novel regulatory phosphoproteins significantly modulated in pre-diabetic and diabetic NOD islets.

experiments, glucose stimulation increased the amount of secreted insulin in a time-dependent manner, with peak insulin secretion occurring after around 30 min (Supplementary Fig. 7A). For subsequent analysis, we focused on around 11,000 sites for which we had quantitative values for at least four of the eight time points (Class 1 sites, Supplementary Data 5). As with our previous phosphoproteome measurements, biological replicates

demonstrated high reproducibility at each time point (Pearson correlation coefficients 0.85–0.95) (Supplementary Fig. 7B). In a PCA, component 1 clearly segregated the control and early (0 and 2.5 min) time points from intermediate (5, 10 and 15 min) and late (20, 30 and 60 min) time points (Fig. 6b). Unsupervised hierarchical clustering of the significantly modulated phospho-peptides (ANOVA, FDR <0.05) revealed four predominant

groups of phosphosites with different dynamic profiles (Supplementary Fig. 7C). Sustained glucose responders were enriched for PKA and PKC substrate motifs, while the transient responders were enriched for AKT and p70S6K kinase substrates, consistent with literature reports[32,33].

Of the 64 phosphosites universally regulated by drug treatment in a glucose-dependent manner, 34 were also significantly regulated by glucose in the time-series studies (Fig. 6c). Discarding two sites that are not conserved between mouse and human, our combined strategy resulted in a high stringency list of 32 potential regulatory phosphosites of which only four had previously been characterized as being functional (Fig. 6d). One of the known sites is S243 of the metabolic enzyme glutamine-fructose-6-phosphate transaminase (Gfpt). Phosphorylation of this residue controls the activity of the enzyme, which so far has been characterized as a regulator of glucose flux into the hexosamine pathway only in the brain[34]. The finding that this site is regulated in the beta cell suggests that this may similarly be an important control point for glucose metabolism in the context of insulin secretion. A large proportion of these phosphosites is involved in the regulation of vesicle trafficking and cell cycle-related functions, providing molecular detail on the connection between insulin secretion and cell proliferation. Among these proteins were two poorly characterized calcium channels, Slc24a2 and Mcoln1, both localized to the endoplasmatic reticulum. We found that overexpression of the unphosphorylable mutants of both of these proteins impairs the ability of the beta cells to secrete insulin upon glucose stimulation (Supplementary Fig. 8). Remarkably the protein level of 10 out of these 32 novel regulatory sites is significantly impaired in NOD pre-diabetic and diabetic islets (Fig. 6e).

**Characterization of the S7 DNMT3A phosphorylation.** Among the new potentially functional phosphosites (Supplementary Fig. 9A), we also identified Serine 4 of the *de novo* DNA methyltransferase DNMT3A (Fig. 6c; Supplementary Fig. 9B,C). This protein plays a crucial role in beta cell differentiation and metabolism. Beta cell-specific deletion of DNMT3A is sufficient to cause beta-to-alpha-cell reprogramming, driving a metabolic program by repressing key genes to enable the coupling of insulin secretion to glucose levels during beta cell maturation[35,36]. Additionally, a genome-wide association study (GWAS) robustly revealed DNMT3A as one of the genetic contributors to the pathogenesis of type 1 diabetes[37]. To investigate the functional role of this phosphorylation site, we employed interaction proteomics of the human protein, where the corresponding site occurs at S7 of DNMT3A.

We coupled immunoprecipitation experiments of the wild-type form of the methyltransferase, as well as the unphosphorylable S7A mutant, to quantitative MS-based proteomics[38,39]. Statistically significant interactors of wild type DNMT3A included a histone deacetylase (HDAC2) and histones (ex. histone H3.1), as expected from the literature[40,41] (Fig. 7a). Remarkably, we found that these interactions were significantly decreased following mutation of this regulated phosphorylation site (Fig. 7b). Both the wild-type form and the unphosphorylable mutant are correctly localized in the nucleus, therefore the differential interaction is not due to aberrant subcellular localization (Supplementary Fig. 10A). We also confirmed the role of S7 phosphorylation in regulating the association of the methyltransferase to its partners in a human cell line, HEK293 (Supplementary Fig. 10B,C).

Recently it has been shown that the histone H3-DNMT3A interaction triggers activation of the methyltransferase after its initial genomic positioning[41]. Given the role of the S7 phosphorylation in modulating the DNMT3A association with its partners (Fig. 7a,b), we decided to investigate its effect on the gene expression profile at the level of the proteome. To minimize potential off-target effects of the overexpression, we performed our experiments in two biological systems, Ins1e and HEK293 cell lines, which express the exogenous Dnmt3a at low and high levels, respectively. In both systems, we ectopically expressed the wild type or the unphosphorylable mutant form of the methyltransferase and compared the proteomes (Supplementary Fig. 10C–G; Supplementary Data 6). Overexpression of wild-type DNMT3A led to downregulation of a set of proteins involved in cell proliferation and insulin signalling regulation (Fig. 7c,g). In contrast, over-expression of the S7A mutant of DNMT3A resulted in drastic upregulation of this set of proteins (mean fold change of more than three), compared with the wild-type form (Fig. 7d). These results, confirmed in the human cell line Hek293 (Supplementary Fig. 10D,E; Supplementary Data 7), demonstrate that the phosphorylation of S7 in response to glucose contributes to the regulation of genomic DNMT3A targets via phosphorylation dependent interactions.

To determine the proportion of gene regulatory events mediated via regulated phosphorylation of S7 of DNMT3A, we compared the proteomes of beta cells stimulated or not by glucose for 6 or 12 h (Supplementary Data 8; Supplementary Fig. 11). Out of 7,000 proteins, about 1,500 were significantly downregulated and this set encompassed 65% of the DNMT3A-suppressed genes (Fig. 7e). Conversely, DNMT3A is involved in 5% of the gene regulatory events induced by glucose. Together, these observations establish that S7 phosphorylation of DNMT3A represents a pathway through which glucose suppresses the expression of an important subset of target genes (Fig. 7f).

## Discussion

Beta cell dysfunction is a major hallmark of the progression of diabetes. Comprehensive identification of the molecular mechanisms triggered by glucose stimulation and governing insulin secretion is thus crucial not only for a deep understanding of this process but also for the characterization of existing and development of novel effective therapeutics targeting this disease.

Here, we applied state of the art, high-resolution MS-based proteomics to quantitatively describe the phosphorylation events occurring in murine beta cells actively secreting insulin. We first profiled the proteomes of NOD mice at different diabetic stages. Then we functionally characterized our experimental system by comparing the proteomic profiles of islets and beta cells to a depth of >9,000 proteins. Apart from validating our cell line system for our purposes, our copy-number estimates of islet proteins will be a useful resource to the community.

We quantified changes in the phosphorylation status of >28,000 Class 1 phosphosites upon glucose stimulation and treatment with diverse drugs affecting insulin secretion. These data classified the insulin secretion-modulating compounds into three groups, representing key nodes in the beta cell signalling network targeted. Through this analysis, we validated known drug–kinase–substrate relationships, and also identified new molecular targets by which compounds regulate insulin secretion. For example, this approach revealed that the casein kinase 2 (CK2) substrate motif is strongly enriched in beta cells treated with Glibenclamide and ATP. Both these drugs increase insulin secretion by augmenting the calcium response[42,43]. The combination of our phosphoproteomic data set with known kinase–substrate relationships, extracted from PhosphositePlus, enabled us to connect the drug-dependent increase in calcium concentration with downstream activation of CK2. Our study suggests that CAMK2 kinase, which is sensitive to intracellular calcium levels, serves as a molecular bridge between the CK2

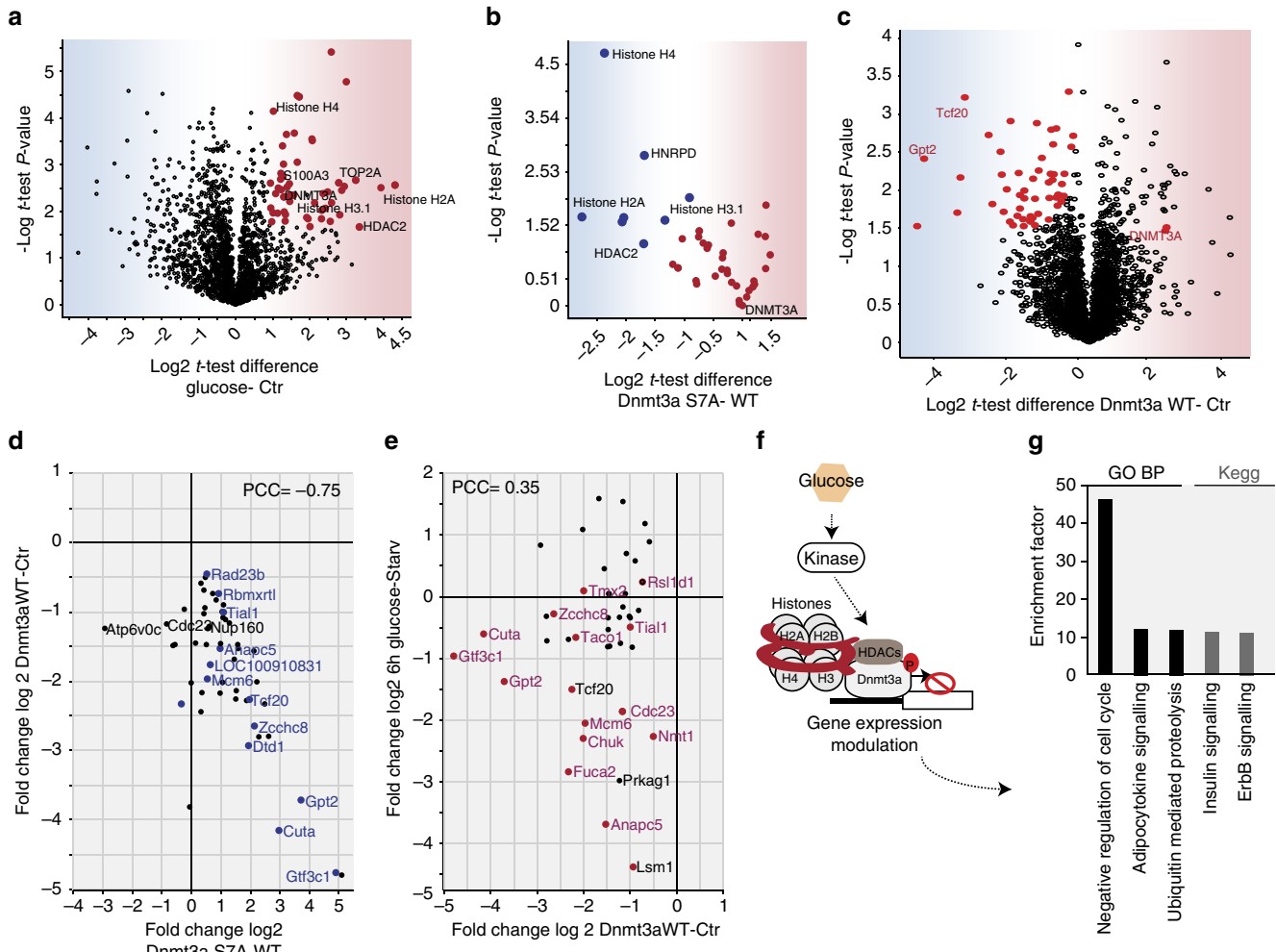

**Figure 7 | The DNMT3A phosphorylation of S7 affects the gene expression profile of beta cells.** (**a**) Volcano plot of MS-quantified wild-type DNMT3A interactors in glucose treated or starved Ins1e cells. (**b**) Significant interactors of DNMT3A. Volcano plot of MS-quantified proteins in wild type and S7A mutant in glucose treated Ins1e cells. (**c**) Volcano plot of MS-quantified proteins after the overexpression of the wild type form of DNMT3A or empty vector as control, in Ins1e cells. (**d**) Scatterplot of proteins significantly downregulated after the overexpression of wild type or S7A mutant forms of DNMT3A in Ins1e cells. (**e**) Scatterplot of significantly downregulated proteins after the overexpression of wild-type form of DNMT3A and stimulation with glucose (16.7 mM) for 6 h in Ins1e cells. (**f**) Schematic representation of the proposed model. (**g**) Bar graph representing GO-Biological processes enriched in the proteins that are downregulated by both glucose and DNMT3A overexpression.

kinase and the treatment with Glibenclamide and ATP. Additionally our approach revealed that ATP triggers the activity of PKC kinase. This observation is in agreement with the current hypothesis proposing that ATP promotes insulin exocytosis partly by raising calcium concentration and partly by increasing DAG via PLC pathway[44,45]. These results illustrate how unbiased MS-based phosphoproteomics can be applied to identify molecular targets involved in known outcomes as well as those involved in as yet uncharacterized effect of the drugs. Mapping our phosphoproteomics data onto signalling networks connected the drugs tested here to cell cycle related and proliferative pathways, which is a desirable effect in view of the fact that type 2 diabetes involves progressive failure of beta cells. Further experiments are necessary to confirm the pro-proliferative properties of these drugs in islets, which have a low proliferation rate compared with Min6 cells.

Our strategy did not reveal a differential regulation of glucose and drugs on phosphatase activity. Although protein phosphatases regulate insulin-secretion pathways, our strategy did not reveal a differential regulation of glucose and drugs on

phosphatase activity[46]. This could be explained by the fact that phosphatases activity is rarely correlated with the phosphorylation level of key specific phosphatase residues[47].

Given the prominent role of autocrine signalling in beta cells, it is important to consider that it is difficult to discriminate whether the mentioned kinases are activated directly and/or indirectly by drugs.

We also generated a quantitative atlas of dynamic protein phosphorylation following glucose stimulation at eight time points. By integrating this large-scale phosphoproteomics data with a manually curated insulin secretion pathway, we delineated key topological features of this signalling network (Fig. 8). Two major pathways regulated by glucose are PI3K-AKT and MAPK, which control nutrient sensing, protein synthesis, metabolism and cell proliferation[48]. In our quantitative phosphoproteome, we observed that in beta cells both these pathways are fully activated upon 10 min of glucose stimulation. The temporal resolution of our time-series study enabled discrimination of AKT and mTOR-p70S6K signalling in the context of regulating insulin, which was delayed with respect to the activation of AKT. This is

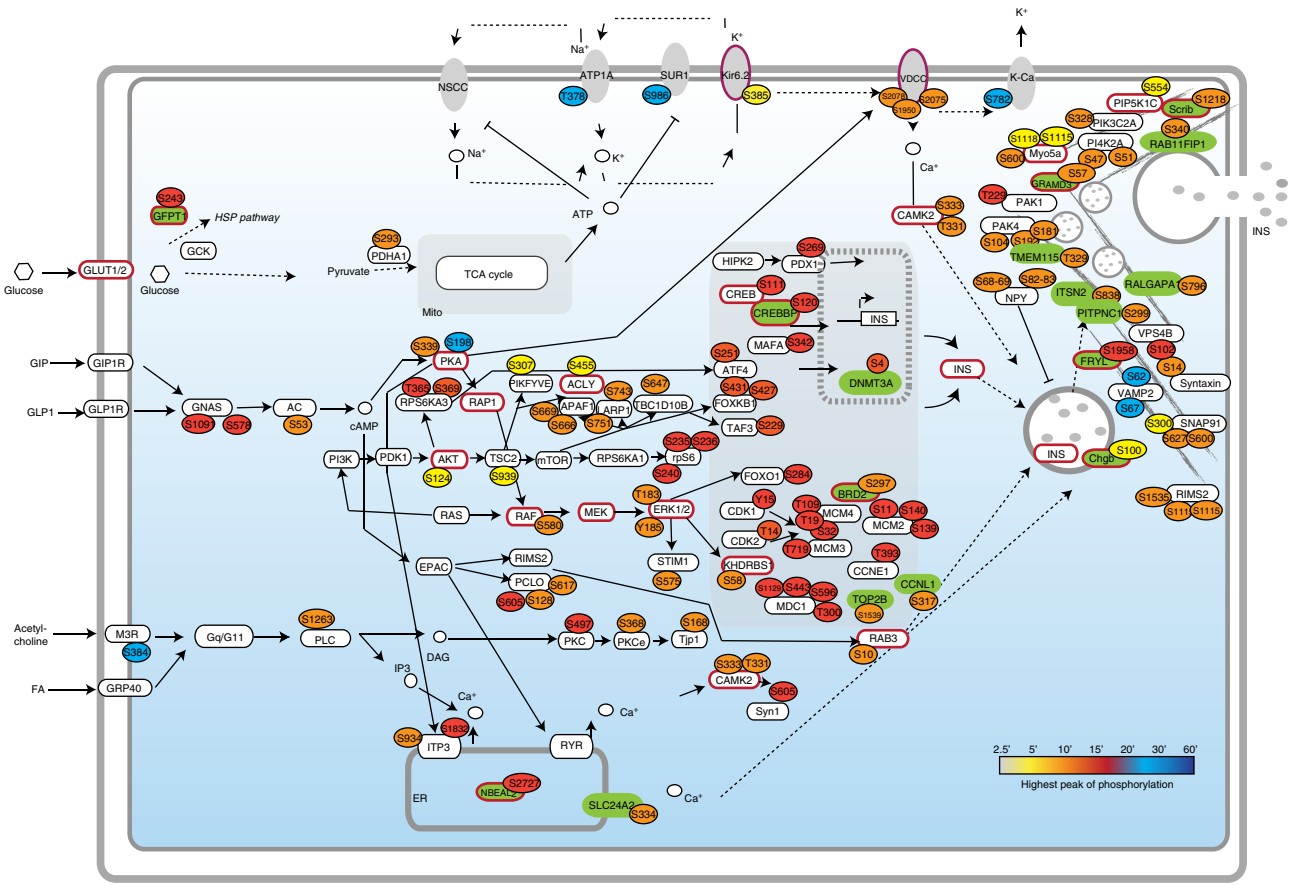

**Figure 8 | Time-resolved map of insulin secretion pathways in beta cells.** Assembly of insulin secretion pathways from multiple databases. Phosphosites are labeled according to their highest phosphorylation peak, as revealed in our phosphoproteomics data set (see legend). New regulatory phosphoproteins are labelled in green. Proteins significantly modulated in diabetic islets have a red border.

consistent with current knowledge of the PI3K-AKT-mTOR signalling network, whereby mTOR is downstream of AKT[49], and the significant temporal latency between the activation of AKT and mTOR[50,51].

Upon 10 min of glucose stimulation almost all proteins involved in the transport and the exocytosis of insulin granules were highly phosphorylated. This observation is in line with established models of insulin secretion, whereby in response to glucose stimulation, a pool of docked and primed insulin storage granules fuse and release insulin, while additional storage granules traffic to join the readily releasable pool at the cell surface[52]. These events are triggered by increased cytoplasmic calcium concentration resulting from plasma membrane depolarization. Consistently we found that upon 10 min of glucose stimulation the voltage-dependent calcium channel was highly phosphorylated and subsequently CAMK2 kinase activated (Fig. 8). These molecular events are elegantly connected to the activation of the PKA kinase, triggered by the generation of cAMP by adenylyl cyclase, which is also highly phosphorylated upon 10 min of glucose stimulation, as revealed by our phosphoproteomics data.

Another crucial feature of the beta cell response to glucose stimulation is the modulation of gene expression profiles through the activation of different transcription factors[53]. We were able to quantitatively measure the phosphorylation profiles of key beta cell transcription factors, such as Pdx1, MafA, ChREBP and Foxo1 and found that 15 min of glucose stimulation is sufficient to trigger their phosphorylation. These data are in line with the EGF-induced phosphorylation of transcription factors, such as

cFos, cJun, Stat5, occurring upon 15 or 20 min of stimulation in cell line systems[54].

Our approach also enabled the identification of new molecular players of the glucose response in murine beta cells. By combining our phosphoproteomics data sets with an *in silico* approach, we identified 32 key phosphorylation events occurring on 29 proteins as high-stringency candidate functional sites in mediating the glucose response in beta cells. Strikingly a significant proportion of these novel regulatory phosphoproteins were downregulated in NOD pre-diabetic and diabetic mice (*P* value < 0.005). This observation further highlights the importance of these proteins in the regulation of insulin secretion. We also identified the DNA methyltransferase, DNMT3A as a new molecular target of glucose-mediated beta cell signalling. This protein plays an important role in beta cells differentiation: DNMT3A binds Nkx2.2, Grg3 and Hdac1 and mediates the repression of the Arx gene, thereby preventing beta to alpha cell conversion[36]. In addition, it has recently been reported that the beta cell–specific deletion of DNMT3A results in loss of the glucose-induced insulin secretion[35]. Here we show that glucose stimulation increases the phosphorylation of S4 of DNMT3A (S7 in human DNMT3A). Follow-up experiments using interaction proteomics revealed that this phosphorylation event mediates the association of the methyltransferase with specific transcription factors and histones. Such interactions are already known to play a crucial role in the regulation of the DNMT3A-mediated repression of its target genes[41]. We now demonstrate that the ability of the methyltransferase to downregulate its targeted genes is regulated through interactions modulated by the

S7A mutation. Taken together, our results indicate that glucose stimulation downregulates a subset of genes involved in the regulation of cell cycle and signalling, through the DNMT3A phosphorylation of S7.

Here we have highlighted only a subset of the glucose-driven physiological processes which we found in our data set. We expect that these large-scale MS-based data will serve as a valuable resource for future hypothesis-driven research to investigate as yet unknown molecular mechanisms driving insulin secretion in pancreatic beta cells, and that studies of this kind will thereby reveal promising new targets for the treatment of diabetes as well as mechanisms of action of diabetic drugs.

## Methods

**Reagents.** Glibenclamide (20 μM), 8-bromo cAMP (10 mM), 8-bromo cGMP (10 mM), SB216763 (40 μM) and ATP (20 μM) were purchased from Sigma. GLP-1 (10 nM) was provided by Sanofi and Carbachol (100 μM) obtained from European Pharmacopeia Reference Standard. Flag-M2 beads were obtained from Sigma (A2220) and Flag-DNMT3A from Invitrogen. Anti-HDAC2 antibody was obtained from Cell Signaling (cat no. 5113, 1:1,000) and anti-Flag antibody from Sigma (cat no. F3165, 1:2,000). Slc24a2 and Mcoln1 constructs from were purchased from Origene.

**Cell culture and transfection.** Min6 cells were cultured in DMEM medium (Glutamax, Gibco) supplemented with 10% fetal calf serum, 100 U ml$^{-1}$ penicillin, 100 μg ml$^{-1}$ streptomycin, 1 mM sodium pyruvate, 1 M Hepes and 50 μM 2-mercaptoethanol. INS-1E cells (kindly provided by Dr Martin Jastroch, Helmholtz center, IDO, Munich) were grown in a humidified atmosphere (5% $CO_2$, 95% air at 37 °C) in monolayer in modified RPMI 1,640 medium supplemented with 10% fetal calf serum, 10 mM Hepes, 100 U ml$^{-1}$ penicillin, 100 μg ml$^{-1}$ streptomycin, 1 mM sodium pyruvate, 50 μM β-mercaptoethanol (all from Gibco) and 0.5% BSA (from Sigma). Hek293 cells were cultured in DMEM medium supplemented with 10% fetal calf serum, 100 U ml$^{-1}$ penicillin, 100 μg ml$^{-1}$ streptomycin. Cells were transfected with Lipofectamine 2,000 or 3,000, according to the manufacture protocol.

**Islet isolation.** Adult C57BL/6 and NOD mice (Jackson Laboratories, ME) were euthanized by cervical dislocation. The upper abdomen was incised to expose liver and intestines. Pancreas was perfused through the common bile duct with cold collagenase P (from Roche) in saline solution. The pancreas was dissected and placed into a warm collagenase saline solution for 15 min. After enzymatic digestion of the pancreatic tissue, islet were picked and cultured overnight in an incubator at 37 °C.

**Insulin assay.** Cells were grown overnight with DMEM low glucose medium, then were washed with Krebs-Ringer-Buffer and incubated with starvation buffer for 90 min. Cells were then incubated with high glucose medium (Krebs-Ringer-Buffer supplemented with glucose 16.7 mM and BSA 0.05%) or low glucose medium (Krebs-Ringer-Buffer supplemented with glucose 2.5 mM and BSA 0.05%). Aliquots of the supernatant were assayed for the amount of insulin (insulin assay from Cisbio), according to the manufacture protocol.

**Immunoprecipitations.** For immunoprecipitations, cells were lysed in ice-cold NP-40 extraction buffer (50 mM Tris-HCl, pH 7.5, 120 mM NaCl, 1 mM EDTA, 6 mM EGTA, 15 mM sodium pyrophosphate and 1% NP-40 supplemented with protease and phosphatase inhibitors (Roche) and clarified by centrifugation at 14,000 r.m. Supernatants were incubated at a concentration of 30 μl of resin per mg lysate over night with Flag-M2 beads (Sigma) previously washed 3 times with PBS. Beads were then washed 3 times with lysis buffer and 3 times with This-HCl 50 mM pH 8.5 On bead digestion of protein complexes used for MS analysis was performed[38,39,55]. Peptides were eluted, desalted and analysed by LC–MS/MS.

**Proteome and phosphoproteome sample preparation.** Cells were lysed in GdmCl buffer. Proteome preparation was done using the in StageTip (iST) method[10]. Large-scale phosphoproteome preparation was performed as previously described[9]. A limited amount of material (1 mg per condition) was lysed, alkylated and reduced in one single step. Then proteins were digested and phosphopetides enriched by $TiO_2$ beads. After elution, samples were separated by HPLC in a single run (without pre-fractionations) and analysed by mass spectrometry.

**Mass spectrometric analyses.** The peptides or phosphopeptides were desalted on StageTips[56] and separated on a reverse phase column (packed in-house with 1.8-μm C18- Reprosil-AQ Pur reversed-phase beads) (Dr Maisch GmbH) over 270 min (single-run proteome and phosphoproteome analysis). Eluting peptides

were electrosprayed and analysed by tandem mass spectrometry on a Q Exactive HF[57,58] (Thermo Fischer Scientific) using HCD based fragmentation, which was set to alternate between a full scan followed by up to five fragmentation scans. Proteome and phosphoproteome data were processed and statistically analysed as described in Supplementary Methods.

**Data availability.** The mass spectrometry proteomics data have been deposited to the ProteomeXchange Consortium (http://proteomecentral.proteomexchange.org/cgi/GetDataset) via the PRIDE[59] partner repository with the data set identifier PXD003850. All other data supporting the findings of this study are available within the article and its supplementary information files or from the corresponding author upon reasonable request.

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

## Acknowledgements

We acknowledge Susanne Kroiss, Igor Paron, Korbinian Mayr and Gabriele Sowa at the MPI for biochemistry and Angelika Sabel, Carina Müller, Manuela Schaffer and Hartmut Moors at Sanofi for their excellent technical support. We thank Martin Steger for fruitful discussions F.S. was supported by EMBO LTF (EMBO ALTF 1555-2012).

## Author contributions

Francesca Sacco, Matthias Schäfer and Matthias Mann conceptualized the study designed the experiments. F.S. performed all the experiments and analysed the data. Francesca Sacco, Sean J. Humphrey and Matthias Mann wrote the paper. All the authors read and approved the manuscript.

## Additional information

**Competing financial interests:** This study was done as a collaboration between the Max-Planck Institute of Biochemistry and Sanofi-Aventis Deutschland (SAD), who provided a grant for this purpose to MPIB. M.M, A.S, T.K & M.S. are employed at Sanofi.

