## [Peer Review File · Nature Communications]

Reviewers' comments:

Reviewer #1 (Remarks to the Author):

In this work the authors focus on cellular response to glucose and other small molecules in a murine cell line Min6. They use a mass spectrometry based approach to analyze the proteome and phosphoproteome related to glucose stimulated signaling events while measuring insulin secretion. Firstly, the authors show that the Min6 cell line resembles beta cells from murine pancreatic islets by a comparative proteome analysis showing overlap in expressed proteins. Next, the authors perform phosphoproteomics on samples treated with low dose glucose (10 min) and high glucose (10 min and 30 min) including seven different compounds with a known influence on insulin secretion. The authors find that compound treatment group and converge on three main pathways involving PKA, PKC and CK2A activation.

The phosphoproteomics is expanded to a temporal profiling of dynamic response to glucose stimulation. The authors narrow down interesting p-sites to 29 by comparing significantly regulated sites from the compound analysis to the expanded glucose time course. Of these, 2 sites (on Slc24a2 and Mcoln1) were validated for their importance to GSIS in Ins1e cells upon overexpression of wild-type and S/A mutant. Additionally, it was shown by interaction proteomics that the known HDAC2 and histone 3.1 interactions with Dnmt3a were dependent on phosphorylation of Dnmt3a S7. Furthermore, wt and S/A mutant Dnmt3a overexpression affected the abundance of several proteins.

The presented work comprises a detailed analysis of signaling events related to glucose stimulated insulin secretion (GSIS). The strength of the study is the depth of the analysis, the application of several compounds and the time course analysis. Although the work is comprehensive, intriguing and identifies novel regulatory events of GSIS, the manuscript in its current format presents with many inconsistencies that need to be addressed by the authors.

Several aspects that need attention as listed below. Particularly, many figures and their reference throughout the results section are not complete. Many additional details need to be added to the Materials and Methods. Furthermore the discussion presents merely as a summary of findings rather than a discussion of the data. The section on p. 10 line 5-9 could serve as a great basis for the discussion.

Major comments

1. To establish the role of Dnmt3a in glucose stimulated insulin secretion the authors overexpress the wild-type or S>A mutant isoforms. Unfortunately, overexpression can have many unintended side effects unrelated to the normal physiological role of the protein. A much better experiment to confirm the role of phosphorylation would be to mutate the phosphorylation site to aspartic acid (S>D) or alanine (S>A) endogenously (e.g., with CRISPR). As it stands the overexpression

of the WT form, by itself, apparently has an impact on the protein expression levels of a large number of proteins; this effect appears to be independent of glucose stimulation or phosphorylation status.

2. The level of overexpression level of Dmmt3a in HEK293 cell is too high. From the LFQ data on the y-axis, it appears that Dmmt3a is expressed at ~8-fold (in log space) higher in the transfected cells compared to the endogenous. I assuming LFQ is in Log2? This information is not provided.

3. The expression level for Dmmt3a should be confirmed by western blot.

4. Some of the critical interacting proteins should also be confirmed by co-IP with the endogenous Dmmt3a, or by proximity ligation assays.

5. There appears to be some confusion between gene expression and protein expression. Changes in gene expression should be measured by quantifying mRNA levels, while changes in protein expression should be measured by quantifying proteins. Essentially all data supporting the link of Dmmt3a S7 phosphorylation to gene expression (figs 6 and S8) rely on analyses that show protein abundance changes. The authors should conduct mRNASeq experiments to quantify gene expression changes, or should select a couple of regulated genes and show that the change in protein expression is also occurring on the mRNA level to support any conclusion based on gene expression.

6. The supplementary tables are confusing in their current presentation. The authors have provided data for each analysis, which is good. However, they need to provide the averages and standard deviations for each set of analyses as well. Moreover, the values presented in many of the tables are undefined. Are these Log2 intensity values?

7. Many of the peptides and proteins have missing values in one or more of the columns, with many peptides having missing values in many of the columns. How were these missing data treated? What was the minimum number of measurements needed to include the peptide? Were n=1 values used for comparisons?

8. Perusing the data in Table S3, it is not obvious that Dmmt3a changes significantly in the different conditions relative to the biological variability within each condition. Perhaps the authors could provide a bar plot of this data with error bars?

9. Because S7 of Dmmt3a is critical to the manuscript, the authors should provide the annotated MS/MS spectra of this peptide, along with the associated spectra from other Dmmt3a phosphorylation sites. This would help the interested reader to assess the accuracy of the site assignments, especially given the 3 different phospho-isoforms of the same peptide, with closely spaced phosphorylation sites.

10. The authors indicate that phosphorylation of Dmmt3a at S7 is involved in GSIS, but fail to show that mutation of this site affects GSIS.

11. What was the rationale behind the selection of Dnmt3a for further analysis?
12. The introduction and summary should state that the work focus on a murine system.
13. Rework the summary to specify the three signaling pathways, which the compound group into and converge on: PKA, PKC, CK2A. Are the three identified pathways important for GSIS? It would be interesting to know if the use of inhibitors for these three kinases reduce GSIS upon treatment?
14. It is not clear from the results section which concentrations of glucose have been used for low dose versus high dose glucose treatments. Please specify the first time the reader is introduced to these terms. Furthermore, there is inconsistency regarding low dose. Was 2.5 mM (p. 21, legend fig 2) or 3.5 mM (p. 18) used? Additionally please specify if high or low dose glucose was used for the expanded time-course in Min6 cells.
15. There are no details on how the murine pancreatic islets were collected for the analyses.
16. The concentrations used for the seven compounds are not included in the manuscript. Furthermore details are missing on how the compounds were used for in the described phosphoproteomics setup such as pretreatment time.
17. There is no reference to figure S1, S7B, 5F, 6E.
18. References to different parts of figure S2 in the text p. 6 do not match the actual figure. Furthermore, the figure legend does not match.
19. Several figures miss the color key (fig 2D, S3, 3D, 5E, S1A), which make it hard to interpret the presented data.
20. Please include kinase-substrate enrichment analysis for GLP-1 and carbachol in Fig S3 to have a valid statement/sentence in text p. 7 line 18.
21. Comments regarding the insulin assays presented in fig 2C , S6A, S7A: It is neither clear from the figures nor materials and methods section whether the data represent mean, median (of how many replicates? Technical or biological?), SEM or SD. Furthermore the text (p. 6 line 3) refers to a peak insulin secretion at 30 min glucose stimulation. This conclusion is not supported since 2C only includes the 30 min time point and no longer time points. Although fig S6A includes a 60 min time point the error bars between 30 min and 60 min overlap. Please note in figure S6A that length of error bars are inconsistent for individual bars. Figure legend to S6A does not state the dose of glucose used (low versus high).
22. p. 8 please specify where the number of "more than 22,000 sites" is derived from? Otherwise please stay consistent with numbers.
23. Figure 3B has to be scaled up to clearly display the motifs.
24. Figure S4 -legend and figure does not match. Furthermore S4B and C is not described in the results.

25. Figure 5S needs reworking. It is difficult to understand.
26. Is a vertical label missing from fig S6C?
27. Why were Ins1e and not Min6 cells chosen for Slc24a and Mcoln1 wt and mutant overexpression? Are wt and mutants expressed to the same level?
28. Is it correct that only histone H3.1 (and not HDAC2) is confirmed in HEK cells? Fig S8B-C.
29. What is the actual overlap in significant protein abundance changes between Ins1E cells and HEK293 cells as presented in fig S8 and fig 6.
30. Is fig 6F referenced correctly in the text?
31. Which cells does the analysis in Table S7 refer to? Min6 cells or Ins1e cells?

Minor comments

1. Fig 1A: numbers in the Venn diagram does not add up and they do not align with the text. Text reports more than 9000 proteins for Min6 but venn shows: $7465+1196=8661$.
2. Supplementary material p. 38: is it correct that the murine data was searched against the human Uniprot FASTA database?
3. Reference to table S1 (should be proteome???) But in p. 43 Table S1 is referring to phospho-data.
4. Figure 4B, please rework the figure so that p-sites are clearly visible.
5. P. 10: 6000 sites or 6041 sites?
6. P. 11 how does 38 phosphosites minus 2 discarded sites end up being 29 sites?

Other things:

Summary:

The sentence "A high-resolution time course revealed key novel regulatory sites and unexpected connections to epigenetic regulation of gene expression through the methyltransferase DNMT3A" is heavily overstated for the last part. There is no direct evidence of gene expression changes related to DNMT3A.

Reviewer #2 (Remarks to the Author):

In this phosphorproteomics study by Sacco, Schaefer, Mann and colleagues, the authors perform an exhaustive array of glucose and multi-drug perturbation/response experiments on an in vitro model of glucose-mediated insulin secretion using Min6 cells and discover specific drug/target response pathways as candidate avenues for therapeutic intervention. Detailed molecular interactomics analyses of the de novo DNA methyltransferase enzyme Dnmt3a revealed glucose-mediated, phospho-dependent functional interactions of this epigenetic regulator with its genomic targets in cells. Type-2 diabetes (T2D) is a major global health burden on society; new therapeutic strategies and approaches are urgently required, which creates significant rationale and potential sustained translational impact for the present work.

This is a well-conceived, thoughtful, careful and technically strong research program and manuscript. Its broad general appeal is also well suited to the readership of Nature Communications. I have only a few minor comments intended to refine and polish what is

already a very strong study. My recommendations are in no particular order:

1. Some additional functional tests or details of the Min6 system would bolster the manuscript. As-is, the validation of these cells as a surrogate for pancreatic islets is described on a strictly proteomics basis. Even if prior published research evaluating the response of Min6 cells to glucose stimulation or other factors is presented, it would bolster the discussion of these results to make them more balanced between descriptive proteomics and functional analysis. Correlation of rank order of protein copy number alone appears insufficient to describe functional equivalency of two quite different systems.
2. Some balance to the discussion regarding the kinase pathways regulated by the three drug groups would soften a few of the untested and unproven assertions in that section. No discussion of the potential differential regulation of these compounds on phosphatase activity, for example, is provided, in spite of the rather strong language regarding the "unambiguous classification" of these drug-response profiles sorting into three kinase-centric signaling pathways.
3. Some additional details of the brief description of how kinase motifs are developed and credentialed would further support the authors' assertions regarding kinase-specific signaling pathways. The three short sentences in the Supplementary Methods are insufficient to reproduce the authors' results.
4. In the discussion of signaling pathways triggered by drug treatment, it is less clear how the authors linked kinase-substrate motifs to pathways that converge on the activation of cell cycle-related kinases, such as Cdks. Some additional details and stepwise transitions through this logic would further clarify these important links.
5. Although it may be beyond the scope of the current work, an ideal analysis of the Dnmt3a phosphosite mutant would include genome-wide methylation array data to assess the direct effects of this phosphorylation site. In addition, the authors could test for rescue of the differential effects of the S7A mutant using a catalytically inactive enzyme. In any case, these data as presented are strongly supportive of the hypothesis that phos-Ser7 is a functional readout of glucose-dependent genomic methylation changes in Min6 cells.

Reviewer #3 (Remarks to the Author):

The work by Sacco et al examines the phosphoproteome of mouse MIN6 insulinoma cells under several conditions that stimulate or augment insulin secretion. The authors have quantified a large number of phospho-sites and show that these cluster into three main signalling groups - depending on the stimulus used. Some novel pathways are also hinted at in this dataset, which should provide a valuable resource and platform for the investigation of insulin secretory function. The authors have largely focused on insulin secretion (as noted in the title for example), although there is some clear indication of an impact of these treatments on other cellular functions such as cycle/apoptosis that should be discussed in the context of the cell model used (i.e. insulinoma cells). Some care should also be taken in interpretation of the results

- for example, does autocrine signalling play a role in some of the observed results? Presumably there should be some conserved signalling between all treatments that increase insulin secretion, since insulin (and other secreted factors) are known to feed-back via cell surface receptors - can this be teased out in the data? Does the data suggest that such feed-back makes only a minor contribution to signalling? Major and minor comments follow:

Major:

1. The authors use several different physiological and pharmacological agents, and it's nice to see in Fig. 2 that the cells appear to respond well to these with respect to their insulin secretion (although note: no units are given in the figure legend or Fig 2C). For the most part, the authors have nicely described the mechanism by which most of the compounds used elicit insulin secretion. However, the description of the effects of extracellularly applied ATP seem not to be entirely correct. The authors lump ATP together with glibenclamide and state that these "...do not directly target a kinases..." (page 7) and that "Both these drugs increase insulin secretion by depolarizing the cell membrane..." (page 14). The reference provided regarding purinergic receptors (Koshimizu et al, 2000) seems out of date and not specific to beta-cells. Although there are some studies implicating P2X receptors in beta-cells, the authors should consider extensive recent literature showing that extracellular ATP (thought to be released from secretory granules to act in an autocrine manner) interacts with G-protein coupled P2Y receptors on beta-cells and is thought to activate PLC-PKC-related pathways as well (for example, Wuttke et al, JBC, 2016; Zhang et al., BBRC, 2015; Khan et al., Diabetologia, 2014; Wuttke et al, FASEB J, 2013; Yelovitch et al, J Med Chem, 2012). As such, the current thinking seems to be that extracellular ATP will increase insulin secretion partly by raising Ca²⁺ (or perhaps more correctly, by 'augmenting' the Ca²⁺ response) and partly by increasing DAG via PLC to promote insulin exocytosis. Is there any evidence for ATP-dependent activation of PLC-PKC (or other) pathways? If not, perhaps this is interesting in itself and merits some discussion.

2. Further to the above point, the authors have very nicely shown that the different agents can be grouped largely according to their presumed mechanism of action. I am somewhat surprised however at how distinctly these groups can be separated. Can the authors comment on cross-talk or overlap between different treatments? Conventional thinking would suggest that autocrine signalling is an important regulatory component of beta-cells (for example, autocrine signalling by several things that might stimulate phosphorylation-dependent signalling within beta-cells are released upon stimulation: insulin, ATP, and GABA are some of these that have been investigated in detail). Would the authors not expect to see some pathways that are always activated by any treatment that promotes secretion - and may be attributed to autocrine activation of signalling pathways (do these represent the 17% overlap mentioned between drugs and glucose-stimulation)? Perhaps this underlies the rationale (which is not well-described in the paper) for using MAPK and PI3K-AKT as indicators of glucose responsiveness (page 7) - since these may be activated by autocrine insulin signalling?

3. In the absence of confirmation in primary islets/beta-cells, I would suggest being cautious about conclusions relating to cell cycle and apoptosis (particularly the former - i.e. Fig. 4) obtained from highly proliferative MIN6 cells. Data from primary islets would significantly strengthen this aspect of the paper (along with the Dnmt3a finding).

4. The majority of the work performed in this study was in MIN6 insulinoma cells. I recognize that the authors have confirmed similarity in protein expression (but not phosphorylation) between these and primary mouse islets, but the authors should still avoid using the term 'beta-cells' throughout the paper when referring to the insulinoma cells (including the title, since the 'insulinoma phosphoproteome' rather than 'beta-cell phosphoproteome' was studied). Some additional examples include: "The finding that this site is regulated in the beta cell..." (page 11); "...we decided to investigate the temporal regulation of these sites and of the global beta cell phosphoproteome in a glucose dependent time-course in beta cells." (page 10).

Minor:

1. The authors describe isolated islets as the "...in vivo cellular context..." of the MIN6 cells. I think that this needs some clarification. Isolated islets certainly do not re-capitulate all of the in-vivo context of beta-cells (i.e. loss of innervation and vascularization". The authors may wish to revise this sentence to reflect that the data from primary islets (which as the authors point out could be a useful resource in itself) in comparison to the MIN6 cells suggests that the latter is indeed a reasonable model to use.

2. The first sentence of the Discussion states that "Beta cell dysfunction is a major hallmark of the progression of type 2 diabetes." The authors may want to consider that the beta cell plays not just an important role in disease progression, but is in fact a major contributor to the initiation and genetic susceptibility to diabetes as well.

3. On page 5, 8-bromo-cGMP is listed twice (one of those should likely be 8-bromo-cAMP).

July 28, 2016

Point-by-point answers to reviewer's comments for "Glucose-regulated and drug perturbed beta-cell phosphoproteome reveals molecular mechanisms controlling insulin secretion" by F. Sacco et al.

We thank all the reviewers for their thoughtful and constructive comments, which were helpful in improving the manuscript.

Reviewer #1 (Remarks to the Author):

In this work the authors focus on cellular response to glucose and other small molecules in a murine cell line Min6. They use a mass spectrometry based approach to analyze the proteome and phosphoproteome related to glucose stimulated signaling events while measuring insulin secretion. Firstly, the authors show that the Min6 cell line resembles beta cells from murine pancreatic islets by a comparative proteome analysis showing overlap in expressed proteins. Next, the authors perform phosphoproteomics on samples treated with low dose glucose (10 min) and high glucose (10 min and 30 min) including seven different compounds with a known influence on insulin secretion. The authors find that compound treatment group and converge on three main pathways involving PKA, PKC and CK2A activation.

The phosphoproteomics is expanded to a temporal profiling of dynamic response to glucose stimulation. The authors narrow down interesting p-sites to 29 by comparing significantly regulated sites from the compound analysis to the expanded glucose time course. Of these, 2 sites (on Slc24a2 and Mcoln1) were validated for their importance to GSIS in Ins1e cells upon overexpression of wild-type and S/A mutant. Additionally, it was shown by interaction proteomics that the known HDAC2 and histone 3.1 interactions with Dnmt3a were dependent on phosphorylation of Dnmt3a S7. Furthermore, wt and S/A mutant Dnmt3a overexpression affected the abundance of several proteins.

The presented work comprises a detailed analysis of signaling events related to glucose stimulated insulin secretion (GSIS). The strength of the study is the depth of the analysis, the application of several compounds and the time course analysis. Although the work is

comprehensive, intriguing and identifies novel regulatory events of GSIS, the manuscript in its current format presents with many inconsistencies that need to be addressed by the authors.

We thank the reviewer for the thorough review and kind comments about our manuscript. We hope the changes and new data we provide further strengthen it in his or her eyes.

Several aspects that need attention as listed below. Particularly, many figures and their reference throughout the results section are not complete. Many additional details need to be added to the Materials and Methods. Furthermore the discussion presents merely as a summary of findings rather than a discussion of the data. The section on p. 10 line 5-9 could serve as a great basis for the discussion.

Major comments

1. To establish the role of Dnmt3a in glucose stimulated insulin secretion the authors overexpress the wild-type or S>A mutant isoforms. Unfortunately, overexpression can have many unintended side effects unrelated to the normal physiological role of the protein. A much better experiment to confirm the role of phosphorylation would be to mutate the phosphorylation site to aspartic acid (S>D) or alanine (S>A) endogenously (e.g., with CRISPR). As it stands the overexpression of the WT form, by itself, apparently has an impact on the protein expression levels of a large number of proteins; this affect appears to be independent of glucose stimulation or phosphorylation status.

We agree with the reviewer that over-expression may have unintended consequences. Therefore, we performed our experiments in two biological systems, Ins1e and HEK293 cell lines, that express the exogenous Dnmt3a at low and high levels, respectively. As shown in Fig. S10 (panel F), while in Ins1e cells the exogenous Dnmt3a level is doubled (increased by a factor of 1 on the log2 scale), in HEK293 cells we observed an 8-fold increase in the Dnmt3a expression. In both systems we demonstrated that the protein expression levels of a large number of proteins is strictly dependent on the phosphorylation of the Ser4 residue of Dnmt3a, as revealed by the comparison between the WT and the non-phosphorylatable S/A mutant of the methyltransferase.

We agree with the reviewer that the mutation of endogenous Dnmt3a using CRISPR technology would be an elegant approach. However, embarking on such experiments would be particularly

time consuming, since the use of CRISPR-Cas9 for the purposes of point mutation is complicated by numerous factors (such as controlling for clonal selection), which currently limit its widespread application. Furthermore, we feel such studies would go beyond the scope of our work, which is not focused on the identification of Dnmt3a-dependent genes themselves, but rather on investigating the functional consequences of Dnmt3a Ser7 phosphorylation. We hope that the reviewer agrees that our unbiased MS-based proteomics and interactomics strategies demonstrate the impact of Dnmt3a on protein expression levels, and that for a large number of proteins this appears to be dependent on Ser4 phosphorylation.

2. The level of overexpression level of Dnmt3a in HEK293 cell is too high. From the LFQ data on the y-axis, it appears that Dnmt3a is expressed at ~8-fold (in log space) higher in the transfected cells compared to the endogenous. I assuming LFQ is in Log2? This information is not provided.

We have now clearly indicated in the text legend and figure S10F that the Dnmt3a LFQ intensity is on a log2 scale. The high level of Dnmt3a expression is consistent with the high transfection efficiency of HEK293 cells.

3. The expression level for Dnmt3a should be confirmed by western blot.

As requested we have now performed western blotting, showing the Dnmt3a over-expression in both Ins1e and HEK293 cell lines. These data are now presented in Figure S10G.

4. Some of the critical interacting proteins should also be confirmed by co-IP with the endogenous Dnmt3a, or by proximity ligation assays.

The most critical and abundant Dnmt3a interactors we found are the HDAC2 and histone H3.1 (Fig. 7). These proteins have already been extensively described as Dnmt3a interactors in the literature ^{1, 2}. We have now added a co-IP experiment, demonstrating that the HDAC2 protein can only bind the wild type form of Dnmt3a, but not the non-phosphorylatable mutant (Fig. S10H). This experiment further confirms our MS-based interactomics results.

5. There appears to be some confusion between gene expression and protein expression. Changes in gene expression should be measured by quantifying mRNA levels, while changes in protein

expression should be measured by quantifying proteins. Essentially all data supporting the link of Dnmt3a S7 phosphorylation to gene expression (figs 6 and S8) rely on analyses that show protein abundance changes. The authors should conduct mRNASeq experiments to quantify gene expression changes, or should select a couple of regulated genes and show that the change in protein expression is also occurring on the mRNA level to support any conclusion based on gene expression.

We thank the reviewer for prompting us to clarify this point. To demonstrate the role of Dnmt3a phosphorylation in the beta cell response to glucose, we have indeed used protein levels as an endpoint rather than mRNA levels. To address the reviewers comments, we took advantage of data recently published by group of Anil Bhushan, who demonstrate a key role of Dnmt3a in initiating a metabolic program essential to preserving the glucose-stimulated insulin secretion (GSIS) capacity during the maturation of beta cells ³. By applying different techniques (from the ChIP assay to bisulfite sequencing analysis of the CpG-rich regions), the authors demonstrate that Dnmt3a represses the expression of the Hk1 and Ldha genes through a direct methylation of their regulatory regions. Additionally, several reports have highlighted the importance of Hk1 and Ldha in the GSIS regulation, since their ectopic expression disrupts the coupling of glucose metabolism to secretion ^{4,5}.

A

B

In line with these results, our MS-based proteomics dataset revealed that after 6 and 12 hours of glucose stimulation, both Hk1 and Ldha genes are significantly down-regulated. We also demonstrated that the over-expression of WT Dnmt3a significantly downregulates the protein level of Hk1 in Ins1e cells. Importantly, the expression levels of these proteins is not affected by the over-expression of the non-phosphorylatable mutant S7A. We hope that the reviewer agrees

that these observations further support the importance of Dmmt3a S7 phosphorylation on gene expression regulation.

6. The supplementary tables are confusing in their current presentation. The authors have provided data for each analysis, which is good. However, they need to provide the averages and standard deviations for each set of analyses as well. Moreover, the values presented in many of the tables are undefined. Are these Log2 intensity values?

We regret any confusion the reviewer has faced with regards to the supplementary tables. To improve presentation, we have now revised supplementary data, and clearly state that the presented values are in Log2 scale. We have now also added averages and standard deviations for each dataset.

7. Many of the peptides and proteins have missing values in one or more of the columns, with many peptides having missing values in many of the columns. How were these missing data treated? What was the minimum number of measurements needed to include the peptide? Were $n=1$ values used for comparisons?

We thank the reviewer for prompting us to clarify this issue. We feel it is important that we provide high quality data in our supplementary materials, and in our phosphoproteome-related tables we therefore only include phosphosites quantified in at least 50% of experimental conditions (12 out of 24). In analyzing the data for this study, we investigated different statistical approaches for reliably identifying glucose- and drug-regulated phosphosites and how to best handle the issue of missing values. Our final choice was to perform the analysis of variance (ANOVA) with FDR control ($FDR < 0.05$) of the Class 1 sites (Table S3 and Table S4), without thresholding the number of quantified values per peptide. The ANOVA test with FDR control inherently penalizes phosphosites that have missing values in many experimental conditions, and such sites are therefore much less likely to be found to be significant. We believe this strikes a good balance between reporting high quality data while at the same time avoiding inadvertently filtering out sites of low abundance that other researchers may find to be biologically important. Below we show the distribution of the valid values in our total phosphoproteome (in blue) and in the ANOVA significant phosphosites (in red). As shown, the ANOVA test identifies the highest

proportion of significant sites among those phosphopeptides quantified in almost all the experimental conditions.

8. Perusing the data in Table S3, it is not obvious that Dnmt3a changes significantly in the different conditions relative to the biological variability within each condition. Perhaps the authors could provide a bar plot of this data with error bars?

As suggested we now provide a plot of the median and SEM of the intensity of the Ser4 phosphorylation in all the different experimental conditions, showing significant variation (ANOVA test, FDR < 0.05). These data can be found in the new Supplementary Figure S9B.

9. Because S7 of Dnmt3a is critical to the manuscript, the authors should provide the annotated MS/MS spectra of this peptide, along with the associated spectra from other Dnmt3a phosphorylation sites. This would help the interested reader to assess the accuracy of the site assignments, especially given the 3 different phospho-isoforms of the same peptide, with closely spaced phosphorylation sites.

We now provide the associated mass spectra of the three different phospho-isoforms of Dnmt3a. These data can be found in the new Supplementary Figure S9C.

10. The authors indicate that phosphorylation of Dnmt3a at S7 is involved in GSIS, but fail to show that mutation of this site affects GSIS.

As the reviewer states, we have not shown that mutation of this site affects the ability of the beta cells to secrete insulin upon glucose stimulation. This is consistent with our MS-based proteomics data, which did not show any significant Dnmt3a-dependent modulation of proteins involved in the regulation of any aspects related to insulin secretion (e.g. vesicle trafficking, exocytosis or membrane hyperpolarization).

Upon Dnmt3a over-expression, we found a set of proteins significantly down-regulated. These proteins are involved in the regulation of the cell cycle and hormone response (insulin and Erbb) and importantly are also significantly down-regulated after 6 hours of glucose stimulation. These data support the importance of Dnmt3a as regulator of a transcriptional program triggered by glucose stimulation.

11. What was the rationale behind the selection of Dnmt3a for further analysis?

Dnmt3a is one of the 31 potential regulatory phosphoproteins involved in the glucose response of beta cells. We selected this protein for further characterization experiments for several reasons: Firstly, it plays a crucial role in beta cell differentiation and metabolism. Beta cell-specific deletion of Dnmt3a is sufficient to cause beta-to-alpha-cell reprogramming, driving a metabolic program by repressing key genes to enable the coupling of insulin secretion to glucose levels during beta cell maturation^{3,6}. Additionally, a genome-wide association study (GWAS) robustly revealed Dnmt3a as one of the genetic contributors to the pathogenesis of type 1 diabetes⁷. We have now edited the text to further clarify the rationale behind the selection of Dnmt3a for our follow-up investigations.

12. The introduction and summary should state that the work focus on a murine system.

We thank the reviewer for this suggestion. We have now edited the introduction and summary text accordingly.

13. Rework the summary to specify the three signaling pathways, which the compound group into and converge on: PKA, PKC, CK2A. Are the three identified pathways important for GSIS? It would be interesting to know if the use of inhibitors for these three kinases reduce GSIS upon treatment?

We thank the reviewer for prompting us to clarify this. As suggested, we have now edited the summary and clarified this point, on page 10 of the results section.

As shown in Figure 3, some of the compounds we used are directly linked to the activation of these three master kinases. As an example: PKA is essential for the GSIS⁸. This data is also supported by the MS-based proteomic profile of NOD diabetic islets we now performed for the revision, in which GSIS is impaired and PKA protein levels are concomitantly reduced (Table S1). We treated beta cells with GLP1, which increases the concentration of cAMP. The GLP1-mediated increase of cAMP as well as the treatment with 8-bromo-cAMP directly activates PKA kinase, potentiating the GSIS. We have also treated beta cells with Carbachol and 8-bromo-cGMP, both triggering PKC/PKG activity. These kinases are also believed to be important for GSIS⁹. ATP and Glibenclamide compounds have so far not been linked to the activation of CK2A, and to our knowledge, our data show for the first time that CK2A is involved in GSIS. Importantly, in agreement with our phosphoproteomics data, the MS-based proteome profiling of NOD diabetic islets also revealed a significant reduction of CK2A levels (Table S1), supporting its implication in GSIS regulation. The combination of our MS-based phosphoproteomics data with the literature-derived signaling network enabled us to reveal that ATP and Glibenclamide likely activate CK2A through CaMK2A activation (Fig. 5B).

14. It is not clear from the results section which concentrations of glucose have been used for low dose versus high dose glucose treatments. Please specify the first time the reader is introduced to these terms. Furthermore, there is inconsistency regarding low dose. Was 2.5 mM (p. 21, legend fig 2) or 3.5 mM (p. 18) used? Additionally please specify if high or low dose glucose was used for the expanded time-course in Min6 cells.

We regret the confusion faced by the reviewer and we have now clarified this in the paper. We have used a high glucose concentration of 16.7 mM and low glucose concentration of 2.5 mM. In

the glucose time-course we have stimulated beta cells for different time points with a high glucose concentration (16.7 mM).

15. There are no details on how the murine pancreatic islets were collected for the analyses.

We have now amended the methods section with details of the experimental procedure we applied to isolate islets.

16. The concentrations used for the seven compounds are not included in the manuscript. Furthermore details are missing on how the compounds were used for in the described phosphoproteomics setup such as pretreatment time.

We thank the reviewer for pointing out this oversight. We have now added the compound concentrations in the methods section of the manuscript.

17. There is no reference to figure S1, S7B, 5F, 6E.

We now include references to these figures in the manuscript.

18. References to different parts of figure S2 in the text p. 6 do not match the actual figure. Furthermore, the figure legend does not match.

We have now edited the text and the figure legend accordingly.

19. Several figures miss the color key (fig 2D, S3, 3D, 5E, S1A), which make it hard to interpret the presented data.

We have now edited these figures to resolve this issue.

20. Please include kinase-substrate enrichment analysis for GLP-1 and carbachol in Fig S3 to have a valid statement/sentence in text p. 7 line 18.

We now include the kinase-substrate enrichment analysis for GLP1 and Carbachol (Figure S4, new panels C and F), and have edited the text accordingly as suggested.

21. Comments regarding the insulin assays presented in fig 2C , S6A, S7A: It is neither clear from the figures nor materials and methods section whether the data represent mean, median (of

how many replicates? Technical or biological?), SEM or SD. Furthermore the text (p. 6 line 3) refers to a peak insulin secretion at 30 min glucose stimulation. This conclusion is not supported since 2C only includes the 30 min time point and no longer time points. Although fig S6A includes a 60 min time point the error bars between 30 min and 60 min overlap. Please note in figure S6A that length of error bars are inconsistent for individual bars. Figure legend to S6A does not state the dose of glucose used (low versus high).

We thank the reviewer for prompting us to clarify this point. In Figure 3C, S7A, S8A we plotted the median of 3-4 biological replicates with the standard deviation (SD). In the text referring to Fig. 2, we state that there is a peak of insulin secretion, compared with the 10-minute stimulation. We chose these two time points for the drug treatment and glucose phosphoproteome profiling in agreement with widely accepted models of GSIS, in which the first peak of insulin secretion occurs after 10 minutes of glucose stimulation, and this is followed by a higher peak of insulin secretion that occurs after 30 minutes of glucose stimulation¹⁰. We have now added a statistical analysis of the insulin secretion assay. These data are now shown in Fig. S7A.

22. p. 8 please specify where the number of "more than 22,000 sites" is derived from? Otherwise please stay consistent with numbers.

We have now clarified in the text that the 22,241 out of 28,637 Class 1 sites (localized with single amino acid resolution) are the phosphosites quantified in at least 50% of our experimental conditions (page 8).

23. Figure 3B has to be scaled up to clearly display the motifs.

We have now scaled the figure as suggested.

24. Figure S4 -legend and figure does not match. Furthermore S4B and C is not described in the results.

We apologize for the confusion and we have now corrected the legends in the Supplementary Information accordingly.

25. Figure 5S needs reworking. It is difficult to understand.

We thank the reviewer for this suggestion and we have now edited the figure to make it clearer.

26. Is a vertical label missing from fig S6C?

We have now added the vertical labeling to Fig. S6C.

27. Why were Ins1e and not Min6 cells chosen for Slc24a and Mcoln1 wt and mutant overexpression? Are wt and mutants expressed to the same level?

Ins1e cells were chosen for the Slc24a2 and Mcoln1 over-expression because Min6 cells are more difficult to transfect compared with Ins1e cells, and because we were also interested in validating our findings in a different beta cell line. We feel this adds credibility to the notion that this may be a general finding of relevance to beta cells. We have now quantified the immunofluorescence signal of anti-FLAG antibody to demonstrate that the WT and mutant forms of Slc24a2 and Mcoln1 are expressed at the same level (Fig. S8C).

28. Is it correct that only histone H3.1 (and not HDAC2) is confirmed in HEK cells? Fig S8B-C.

In our MS-based interactomics experiments we were not able to detect the HDAC2 proteins. However, in the new co-IP experiments (Fig. S10H) between the HDAC2 and Dnmt3a WT and S7A proteins, we confirmed the result previously obtained in Ins1e cells (Fig. 6).

29. What is the actual overlap in significant protein abundance changes between Ins1E cells and HEK293 cells as presented in Fig. S8 and Fig. 6.

In this figure we have represented the actual overlap between the HEK293 and Ins1e cells. Among the 2,007 proteins that we found to be expressed in both HEK293 and Ins1e cells (after homology mapping using their gene names), and among these 133 are up or down regulated in HEK293 and 141 in Ins1e cells. 9 are significantly up or down-regulated in both these two cellular systems and this overlap is statistically significant ($p= 4.5 \cdot 10^{-10}$). The small overlap between these two systems is not surprising given the profound differences of the two cell lines HEK293 (Human Embryonic Kidney cells), versus Ins1e (Rat Insulinoma cells), and the different levels of Dnmt3a expression observed.

9 are significantly either up or down-regulated in both Hek293 and Ins1e cells

Gene names	HEK 293			Ins1e		
	N: -Log Student's T-test value	N: Student's T-test Difference	N: Student's T-test statistic	N: -Log Student's T-test value	N: Student's T-test Difference	N: Student's T-test statistic
	WT_NT	WT_NT	WT_NT	WT_Ins	WT_Ins	WT_Ins
GLO1	2.08219	-0.427833	-4.86058	1.3377	-1.0898	-2.85953
GPT2	1.70605	-0.299	-3.76576	1.53425	-2.92543	-3.32562
PITPNA	1.41481	-0.224833	-3.03804	1.68575	-0.303733	-3.71199
DHX9	1.44266	-0.158199	-3.10385	1.5843	-0.7316	-3.45047
LAMTOR5	1.48189	0.226967	3.19782	1.75065	1.69572	3.88564
HDGFRP2	1.53679	0.249767	3.3319	1.50916	-1.73931	-3.26404
RARS	2.28911	0.271901	5.55526	1.3082	0.187667	2.7926
RPL5	1.70843	0.3857	3.77209	1.31999	1.31513	2.81927
GABARAPL2	3.11895	3.85897	9.24687	1.49468	1.10085	3.22878

30. Is fig 6F referenced correctly in the text?

We thank the reviewer for pointing this. We have revised the text, adding the correct reference to the figure.

31. Which cells does the analysis in Table S7 refer to? Min6 cells or Ins1e cells?

Table S7 contains the Min6 proteomics dataset. We have now clarified this in the legend.

Minor comments

1. Fig 1A: numbers in the Venn diagram does not add up and they do not align with the text. Text reports more than 9000 proteins for Min6 but venn shows: 7465+1196= 8661.

More than 9,000 proteins were identified and 8,661 were quantified in Min6 cells. We have revised the text accordingly.

2. Supplementary material p. 38: is it correct that the murine data was searched against the human Uniprot FASTA database?

We thank the reviewer for pointing out this mistake. Murine data were searched against the Mus Musculus Uniprot FASTA database. We have corrected the supplementary material accordingly.

3. Reference to table S1 (should be proteome???) But in p. 43 Table S1 is referring to phospho-data.

We have now edited the text accordingly, correcting the legend.

4. Figure 4B, please rework the figure so that p-sites are clearly visible.

We have now edited the figure according to the reviewer's suggestion.

5. P. 10: 6000 sites or 6041 sites?

The ANOVA significant phosphosites are 6,041. We have clarified this in the text.

6. P. 11 how does 38 phosphosites minus 2 discarded sites end up being 29 sites?

We apologize for the confusion and we thank the reviewer for prompting us to look again at our tables. 35 phosphosites are confirmed in the time-course experiment; two are discarded and we end up with 33 potential regulatory sites. We have corrected this in the manuscript.

Other things:

Summary:

The sentence "A high-resolution time course revealed key novel regulatory sites and unexpected connections to epigenetic regulation of gene expression through the methyltransferase DNMT3A" is heavily overstated for the last part. There is no direct evidence of gene expression changes related to DNMT3A.

We have revised the summary according to the reviewer's suggestion.

Reviewer #2 (Remarks to the Author):

In this phosphorproteomics study by Sacco, Schaefer, Mann and colleagues, the authors perform an exhaustive array of glucose and multi-drug perturbation/response experiments on an in vitro model of glucose-mediated insulin secretion using Min6 cells and discover specific drug/target response pathways as candidate avenues for therapeutic intervention. Detailed molecular interactomics analyses of the de novo DNA methyltransferase enzyme Dnmt3a revealed glucose-mediated, phospho-dependent functional interactions of this epigenetic regulator with its genomic targets in cells. Type-2 diabetes (T2D) is a major global health burden on society; new therapeutic strategies and approaches are urgently required, which creates significant rationale and potential sustained translational impact for the present work.

This is a well-conceived, thoughtful, careful and technically strong research program and manuscript. Its broad general appeal is also well suited to the readership of Nature Communications. I have only a few minor comments intended to refine and polish what is already a very strong study.

We thank the reviewer for the thorough review and kind comments about our manuscript, and especially for pointing out its broad general appeal. We hope that the changes we have made in response to his or her comments further strengthen this view of our manuscript.

My recommendations are in no particular order:

1. Some additional functional tests or details of the Min6 system would bolster the manuscript. As-is, the validation of these cells as a surrogate for pancreatic islets is described on a strictly proteomics basis. Even if prior published research evaluating the response of Min6 cells to glucose stimulation or other factors is presented, it would bolster the discussion of these results to make them more balanced between descriptive proteomics and functional analysis. Correlation of rank order of protein copy number alone appears insufficient to describe functional equivalency of two quite different systems.

We agree that the correlation of rank order of protein copy number alone is insufficient to infer functional equivalency of Min6 cells and islets. We have now revised the text, highlighting that Min6 cells were chosen primarily because of their ability to recapitulate the release of insulin

after glucose stimulation. Importantly, the Min6 experimental system we used recapitulates the glucose stimulated insulin secretion (GSIS) that occurs in islets: we detected a first peak of insulin secretion after 10 minutes of glucose stimulation, and a second peak after 30 minutes of glucose stimulation (Fig. 2B and S7A). We hope that this functional data together with the comparable expression level of key beta cells proteins, together with the substantial literature, demonstrates to the reviewer that this system is a suitable model for studying some of the signaling processes underlying GSIS.

2. Some balance to the discussion regarding the kinase pathways regulated by the three drug groups would soften a few of the untested and unproven assertions in that section. No discussion of the potential differential regulation of these compounds on phosphatase activity, for example, is provided, in spite of the rather strong language regarding the "unambiguous classification" of these drug-response profiles sorting into three kinase-centric signaling pathways.

We thank the reviewer for this suggestion, and have now revised the discussion text avoiding overstatements.

3. Some additional details of the brief description of how kinase motifs are developed and credentialed would further support the authors' assertions regarding kinase-specific signaling pathways. The three short sentences in the Supplementary Methods are insufficient to reproduce the authors' results.

Prompted by the reviewer we have now added a more detailed description of the kinase-substrate motif analysis we performed. The kinase substrate motifs were extracted from the HPRD database¹¹ and added as annotation to each phosphosite quantified in our phosphoproteome dataset. We subsequently performed two different analyses:

- Fisher exact test of the loadings (phosphosites) responsible for the discrimination of the three groups (Fig. 3B).
- 1D annotation enrichment analyses to identify statistically significant enriched kinase-substrates motifs in the various experimental conditions (Fig. 4A and S3).

We have now edited the Supplementary Methods to include this additional information.

4. In the discussion of signaling pathways triggered by drug treatment, it is less clear how the authors linked kinase-substrate motifs to pathways that converge on the activation of cell cycle-related kinases, such as Cdks. Some additional details and stepwise transitions through this logic would further clarify these important links.

We thank the reviewer for prompting us to clarify this point. The 1D annotation analysis of the kinase substrate motifs in the different experimental conditions (Fig. 4A) revealed a significant enrichment of CDK motifs after treatment with high glucose as well as with all the seven drugs we used. This data is in agreement with the results we obtained by mapping our phosphoproteomics dataset with kinase-substrate networks extracted from the PhosphoSitePlus database¹². This strategy enables us to “walk” through the signal transduction events triggered by drug treatments, revealing that each drug activates different pathways converging upon activation of cell-cycle related kinases, such as Cdk1, Cdk2, Cdk5 and Cdk7. We have now revised the paper to clarify this.

5. Although it may be beyond the scope of the current work, an ideal analysis of the Dnmt3a phosphosite mutant would include genome-wide methylation array data to assess the direct effects of this phosphorylation site. In addition, the authors could test for rescue of the differential effects of the S7A mutant using a catalytically inactive enzyme. In any case, these data as presented are strongly supportive of the hypothesis that phos-Ser7 is a functional readout of glucose-dependent genomic methylation changes in Min6 cells.

We agree with the reviewer that the analysis of the genome-wide methylation with the Dnmt3a phosphosite mutant would be extremely interesting, but beyond the scope of the current work. We thank the reviewer for his or her comment about our follow-up experiments demonstrating the functional role of the phospho Ser7 of Dnmt3a.

Reviewer #3 (Remarks to the Author):

The work by Sacco et al examines the phosphoproteome of mouse MIN6 insulinoma cells under several conditions that stimulate or augment insulin secretion. The authors have quantified a large number of phospho-sites and show that these cluster into three main signalling groups - depending on the stimulus used. Some novel pathways are also hinted at in this dataset, which should provide a valuable resource and platform for the investigation of insulin secretory function. The authors have largely focused on insulin secretion (as noted in the title for example), although there is some clear indication of an impact of these treatments on other cellular functions such as cycle/apoptosis that should be discussed in the context of the cell model used (i.e. insulinoma cells). Some care should also be taken in interpretation of the results - for example, does autocrine signalling play a role in some of the observed results? Presumably there should be some conserved signalling between all treatments that increase insulin secretion, since insulin (and other secreted factors) are known to feed-back via cell surface receptors - can this be teased out in the data? Does the data suggest that such feed-back makes only a minor contribution to signalling?

Major and minor comments follow:

Major:

1. The authors use several different physiological and pharmacological agents, and it's nice to see in Fig. 2 that the cells appear to respond well to these with respect to their insulin secretion (although note: no units are given in the figure legend or Fig 2C). For the most part, the authors have nicely described the mechanism by which most of the compounds used elicit insulin secretion. However, the description of the effects of extracellularly applied ATP seem not to be entirely correct. The authors lump ATP together with glibenclamide and state that these "...do not directly target a kinases..." (page 7) and that "Both these drugs increase insulin secretion by depolarizing the cell membrane..." (page 14). The reference provided regarding purinergic receptors (Koshimizu et al, 2000) seems out of date and not specific to beta-cells. Although there are some studies implicating P2X receptors in beta-cells, the authors should consider extensive recent literature showing that extracellular ATP (thought to be released from secretory granules

to act in an autocrine manner) interacts with G-protein coupled P2Y receptors on beta-cells and is thought to activate PLC-PKC-related pathways as well (for example, Wuttke et al, JBC, 2016; Zhang et al., BBRC, 2015; Khan et al., Diabetologia, 2014; Wuttke et al, FASEB J, 2013; Yelovitch et al, J Med Chem, 2012). As such, the current thinking seems to be that extracellular ATP will increase insulin secretion partly by raising Ca²⁺ (or perhaps more correctly, by 'augmenting' the Ca²⁺ response) and partly by increasing DAG via PLC to promote insulin exocytosis. Is there any evidence for ATP-dependent activation of PLC-PKC (or other) pathways? If not, perhaps this is interesting in itself and merits some discussion.

We thank the reviewer for this comment. In figure 4B, we have shown the main signal transduction pathways triggered by the different drugs. Specifically, ATP treatment is connected to an increased intracellular calcium concentration, which in turn activates CaMK2A and CK2A, and to increased activity of PKC kinase, as revealed by the phosphorylation level of its regulatory serine residue. We have now revised the discussion and updated the references, highlighting that our data are in agreement with the accepted mechanism of action of ATP, proposing that this compound increases insulin secretion partly by triggering the Ca²⁺ response and partly by increasing DAG via PLC and PKC to promote insulin exocytosis.

2. Further to the above point, the authors have very nicely shown that the different agents can be grouped largely according to their presumed mechanism of action. I am somewhat surprised however at how distinctly these groups can be separated. Can the authors comment on cross-talk or overlap between different treatments? Conventional thinking would suggest that autocrine signalling is an important regulatory component of beta-cells (for example, autocrine signalling by several things that might stimulate phosphorylation-dependent signalling within beta-cells are released upon stimulation: insulin, ATP, and GABA are some of these that have been investigated in detail). Would the authors not expect to see some pathways that are always activated by any treatment that promotes secretion - and may be attributed to autocrine activation of signalling pathways (do these represent the 17% overlap mentioned between drugs and glucose-stimulation)? Perhaps this underlies the rationale (which is not well-described in the paper) for using MAPK and PI3K-AKT as indicators of glucose responsiveness (page 7) - since these may be activated by autocrine insulin signalling?

As the reviewer states, the principal component analysis of the phosphoproteomes measured with each treatment condition segregates the drugs into three distinct main clusters. These three groups are enriched in three major kinases reflecting the presumed mechanism of action of each compound. By mapping our based-phosphoproteomics dataset with kinase-substrate network, extracted by the PhosphoSitePlus database ¹², we found that drug treatments overlap in the modulation of different signaling pathways (eg. MAPK1/2 are activated by ATP, glibenclamide, carbachol and 8-bromo-cGMP; while AKT is activated by treatment with all the drugs). This overlap could be explained by the autocrine signaling mentioned by the reviewer, although we would then expect that many proteins involved in the insulin signaling (e.g. RAF-MAPK axis and PI3K-mTOR axis) should be activated by the treatment with all the drugs. As shown in Figure 4B, while the MAPK cascade is differentially modulated by the different drugs, the AKT-mTOR pathway appears to be perturbed by the stimulation with all of the drugs. In addition, drug effects on pathway modulation may overlap because of the highly interconnected nature of signaling networks, and the large number of proteins modulated by the three major kinases (PKA, PKC and CK2A).

We have now revised the discussion, highlighting that from our dataset it is difficult to discriminate signaling events directly due to drug treatment or indirectly to autocrine signaling events. The 17% of sites mentioned by the reviewer are the ones significantly regulated by at least one of the drugs (ANOVA, FDR<0.05). Therefore, we do not think that their phosphorylation may be attributed to autocrine activation. We have used the GSIS as well as the MAPK and mTOR activation to check that the glucose stimulation was working as expected. We have now revised the text highlighting that that the MAPK and mTOR pathways can be activated by autocrine signaling or by calcium and cAMP dependent mechanisms ¹³.

3. In the absence of confirmation in primary islets/beta-cells, I would suggest being cautious about conclusions relating to cell cycle and apoptosis (particularly the former - i.e. Fig. 4) obtained from highly proliferative MIN6 cells. Data from primary islets would significantly strengthen this aspect of the paper (along with the Dnmt3a finding).

We agree with the reviewer and we have now revised the discussion, highlighting that one of the most important difference between the islets and Min6 cells concerns the cell proliferation

control. We have also edited the text avoiding over-statements regarding the cell cycle and apoptosis-related effects we observed upon drug treatments.

4. The majority of the work performed in this study was in MIN6 insulinoma cells. I recognize that the authors have confirmed similarity in protein expression (but not phosphorylation) between these and primary mouse islets, but the authors should still avoid using the term 'beta-cells' throughout the paper when referring to the insulinoma cells (including the title, since the 'insulinoma phosphoproteome' rather than 'beta-cell phosphoproteome' was studied). Some additional examples include: "The finding that this site is regulated in the beta cell..." (page 11); "...we decided to investigate the temporal regulation of these sites and of the global beta cell phosphoproteome in a glucose dependent time-course in beta cells." (page 10).

We agree with the reviewer and have added a statement to the second paragraph of the results section, highlighting that Min6 cells are an insulinoma cell line. We think that it is important to consider that Min6 cells are able to secrete insulin after glucose stimulation, which is the key beta-cell relevant feature that we are focusing on in this study. Since our paper aims at the identification of functional mechanisms underlying glucose induced insulin secretion, we would like to keep the "beta cells" instead of "insulinoma" cells in the title. We feel this makes it clearer to readers that our manuscript is focused on the GSIS aspect of beta cell signaling, rather than on cancer-related biology.

Minor:

1. The authors describe isolated islets as the "...in vivo cellular context..." of the MIN6 cells. I think that this needs some clarification. Isolated islets certainly do not re-capitulate all of the in-vivo context of beta-cells (i.e. loss of innervation and vascularization". The authors may wish to revise this sentence to reflect that the data from primary islets (which as the authors point out could be a useful resource in itself) in comparison to the MIN6 cells suggests that the latter is indeed a reasonable model to use.

We agree with the reviewer and we have rephrased this sentence as suggested.

2. The first sentence of the Discussion states that "Beta cell dysfunction is a major hallmark of the progression of type 2 diabetes." The authors may want to consider that the beta cell plays not just an important role in disease progression, but is in fact a major contributor to the initiation and genetic susceptibility to diabetes as well.

We thank the reviewer for this comment. We have revised the sentence as suggested.

3. On page 5, 8-bromo-cGMP is listed twice (one of those should likely be 8-bromo-cAMP).

We have corrected this mistake.

References

1. Fuks F, Burgers WA, Godin N, Kasai M, Kouzarides T. Dnmt3a binds deacetylases and is recruited by a sequence-specific repressor to silence transcription. *The EMBO journal* **20**, 2536-2544 (2001).
2. Guo X, *et al.* Structural insight into autoinhibition and histone H3-induced activation of DNMT3A. *Nature* **517**, 640-644 (2015).
3. Dhawan S, *et al.* DNA methylation directs functional maturation of pancreatic beta cells. *The Journal of clinical investigation* **125**, 2851-2860 (2015).
4. Alcazar O, Tiedge M, Lenzen S. Importance of lactate dehydrogenase for the regulation of glycolytic flux and insulin secretion in insulin-producing cells. *The Biochemical journal* **352 Pt 2**, 373-380 (2000).
5. Becker TC, BeltrandelRio H, Noel RJ, Johnson JH, Newgard CB. Overexpression of hexokinase I in isolated islets of Langerhans via recombinant adenovirus. Enhancement of glucose metabolism and insulin secretion at basal but not stimulatory glucose levels. *The Journal of biological chemistry* **269**, 21234-21238 (1994).
6. Papizan JB, *et al.* Nkx2.2 repressor complex regulates islet beta-cell specification and prevents beta-to-alpha-cell reprogramming. *Genes & development* **25**, 2291-2305 (2011).
7. Bradfield JP, *et al.* A genome-wide meta-analysis of six type 1 diabetes cohorts identifies multiple associated loci. *PLoS genetics* **7**, e1002293 (2011).
8. Drucker DJ. The biology of incretin hormones. *Cell metabolism* **3**, 153-165 (2006).

9. Schmitz-Peiffer C, Biden TJ. Protein kinase C function in muscle, liver, and beta-cells and its therapeutic implications for type 2 diabetes. *Diabetes* **57**, 1774-1783 (2008).
10. Fu Z, Gilbert ER, Liu D. Regulation of insulin synthesis and secretion and pancreatic Beta-cell dysfunction in diabetes. *Current diabetes reviews* **9**, 25-53 (2013).
11. Keshava Prasad TS, *et al.* Human Protein Reference Database--2009 update. *Nucleic acids research* **37**, D767-772 (2009).
12. Hornbeck PV, Zhang B, Murray B, Kornhauser JM, Latham V, Skrzypek E. PhosphoSitePlus, 2014: mutations, PTMs and recalibrations. *Nucleic acids research* **43**, D512-520 (2015).
13. Briaud I, Lingohr MK, Dickson LM, Wrede CE, Rhodes CJ. Differential activation mechanisms of Erk-1/2 and p70(S6K) by glucose in pancreatic beta-cells. *Diabetes* **52**, 974-983 (2003).

Reviewers' Comments:

Reviewer #1 (Remarks to the Author):

The authors have addressed many of the concerns that were raised in the first round of review either experimentally but mostly through written comments.

Also many of the figures have been reworked or polished and are now presented in a reasonable form, which allows for a proper interpretation. Despite these efforts the manuscript still needs some reworking. There is no doubt about the overall quality of the work and the importance for the insulin community. However, a fair amount of inconsistencies need to be addressed in order for the work to be strong, concise and convincing.

Issues Remaining:

Although the authors have attempted to rationalize their use of “gene expression profile”, the fact remains that gene expression was never actually measured in this manuscript. To minimize confusion for the term “gene expression profile” needs to be removed from legend titles and section subtitles to ensure a precise interpretation of the work presented.

The level of Dnmt3a over-expression in HEK293 cells remains a major issue. The authors point out in the response letter that the LFQ axis is log₂ based. An 8-fold change in log₂ space (Figure S10F) is a 256-fold increase in protein expression! This super-physiological level of expression can have many unintended consequences, especially for a transcriptional regulator. The role of phosphorylation on this massively overexpressed protein might be very different from the same protein expressed at endogenous levels. These experiments need to be repeated with a lower level of transfection and therefore decreased overexpression.

The validation by co-IP of Dnmt3a and HDAC2 (Fig S10H) in the HEK293 cells is weak, at best. Even with the massive overexpression of Dnmt3a, only a barely detectable amount of HDAC2 is being pulled down, despite a high level of expression of HDAC2 in these cells, as apparent from the amount in the WCL fraction. Presumably, if this interaction were real and phospho-dependent, there would be a significant detectable difference in these lanes and HDAC2 would have been detected in the MS analysis.

There seems to be some confusion regarding numbers of identified and quantified proteins in islets and Min6 cells (Table S2) and text p. 8 and figure 2A (venn diagram). Please ensure coherence in the presentation of data. Table S2 reads a total of 9063 rows corresponding to total quantified proteins. Consulting the LFQ columns (Table S2) it seems like in Min6 cells 8466 proteins were quantified and in islets 7555.

Please be clear and concise as to numbers referring to “identified” or “identified and quantified”.

Table S4 presents the numbers of ANOVA significant phosphosites and repeatedly 6041 is reported throughout the manuscript. However, consulting the table it has 6042 rows – 2 header row=6040 phosphosites. Please correct this error.

Fig 4D: Please reassure that numbers listed in text, figure and legend are in agreement. (22,241 vs 22,547).

Text p. 13: There is still some confusion regarding number of regulatory sites. In the response to reviewers the number was corrected to 33. The text still reads 31 or 30 (p. 13 and p. 14, p. 19 reads 30?). When you count the number of listed p-sites in the figure it reads 32. Please readdress this issue.

There are 3 legends to supplementary figure S8. Please correct numbering.

Fig S1B referenced in the text p. 5 reads 0.85-0.99. From the figure it reads 0.68-0.93. Please ensure coherence.

Fig S1C it is not clear from the figure, legend or text what is z-scored. Ratios? Log₂ LFQ?

FigS3B, C is referenced in the text p. 8 to support a “high degree of overlap” between experimental conditions. Please clarify or show support of this statement.

The magnitude of regulation should be made clear by adding numbers to color keys in fig. 3D and 4E.

A vertical title is missing from fig S5B and as previously mentioned also still for fig S7C.

The reference to figures in the last section of the results does not match. (p.15-16).

Please make sure that the correct table S4/S5 is referenced on p. 13.

In the discussion Figure 7 is referenced several times. Please check that this is correct.

Discussion p. 17 please check section 2 for sentence repetitions.

Fig S11 is not referenced and the color key seems off scale.

Text p. 7, one compound is still listed twice.

Text p. 8, what is the calculation behind the statement 65% of the detected proteome?

p. 15 please specify in the text what “we also validated the interaction results in a human cell line, HEK293” entails. Which or how many proteins were confirmed?

Reviewer #2 (Remarks to the Author):

All of my concerns have been thoroughly addressed in revision.

Reviewer #3 (Remarks to the Author):

The authors present a very nice revised paper and have adequately addressed my previous concerns, with one exception:

The authors maintain that it is appropriate to refer to MIN6 cells as 'beta-cells' so as to 'emphasize that the work focuses on the GSIS aspect of these cells' (this is a good point I think). However, I disagree with this. MIN6 cells are not beta-cells and to refer to them as such is a bit misleading (one could argue that they are not even the best insulinoma model, but that's beside the point). If the authors wish to emphasize the focus on GSIS, why not refer to "...the drug perturbed phosphoproteome in insulin-secreting cells..."?

September 5, 2016

Point-by-point answers to reviewer's comments for "Glucose-regulated and drug perturbed beta-cell phosphoproteome reveals molecular mechanisms controlling insulin secretion" by F. Sacco et al.

We thank all the reviewers for their positive comments.

Reviewer #1 (Remarks to the Author):

The authors have addressed many of the concerns that were raised in the first round of review either experimentally but mostly through written comments.

Also many of the figures have been reworked or polished and are now presented in a reasonable form, which allows for a proper interpretation. Despite these efforts the manuscript still needs some reworking. There is no doubt about the overall quality of the work and the importance for the insulin community. However, a fair amount of inconsistencies need to be addressed in order for the work to be strong, concise and convincing.

We thank the reviewer for the positive comments about our manuscript. We have now edited the manuscript text according the reviewer and editor suggestions.

Issues Remaining:

Although the authors have attempted to rationalize their use of "gene expression profile", the fact remains that gene expression was never actually measured in this manuscript. To minimize confusion for the term "gene expression profile" needs to be removed from legend titles and section subtitles to ensure a precise interpretation of the work presented.

We agree with the reviewer that we didn't provide any measurement of gene expression profile. We have now revised the text, explaining that we have used the MS-based proteomic data as a proxy of the gene expression profile.

The level of Dnmt3a over-expression in HEK293 cells remains a major issue. The authors point out in the response letter that the LFQ axis is log₂ based. An 8-fold change in log₂ space (Figure S10F) is a 256-fold increase in protein expression! This super-physiological level of expression can have many unintended consequences, especially for a transcriptional regulator. The role of phosphorylation on this massively overexpressed protein might be very different from the same protein expressed at endogenous levels. These experiments need to be repeated with a lower level of transfection and therefore decreased overexpression.

As the editor suggested, we have now revised text, acknowledging the caveats that may come with over-expression.

The validation by co-IP of Dnmt3a and HDAC2 (Fig S10H) in the HEK293 cells is weak, at best. Even with the massive overexpression of Dnmt3a, only a barely detectable amount of HDAC2 is being pulled down, despite a high level of expression of HDAC2 in these cells, as apparent from the amount in the WCL fraction. Presumably, if this interaction were real and phospho-dependent, there would be a significant detectable difference in these lanes and HDAC2 would have been detected in the MS analysis.

We thank the reviewer for prompting us to clarify this issue in our manuscript. The interaction between HDAC2 and Dnmt3a has been already described by Fuks et al. ¹. In addition we also observed that this interaction is phospho-dependent in our MS analysis in Ins1e cells (Fig 7B). As indicated by the reviewer, in Hek293 cells this interaction is weak. This result is also consistent with the fact that we were not able to detect this association in Hek293 by MS analysis.

There seems to be some confusion regarding numbers of identified and quantified proteins in islets and Min6 cells (Table S2) and text p. 8 and figure 2A (venn diagram). Please ensure coherence in the presentation of data. Table S2 reads a total of 9063 rows corresponding to total quantified proteins. Consulting the LFQ columns (Table S2) it seems like in Min6 cells 8466 proteins were quantified and in islets 7555.

Please be clear and concise as to numbers referring to “identified” or “identified and quantified”.

We thank the reviewer for pointing out this oversight. We have now revised the text and Figure 2A accordingly.

Table S4 presents the numbers of ANOVA significant phosphosites and repeatedly 6041 is reported throughout the manuscript. However, consulting the table it has 6042 rows – 2 header row=6040 phosphosites. Please correct this error.

We thank the reviewer for pointing out this oversight. We have now revised the text accordingly.

Fig 4D: Please reassure that numbers listed in text, figure and legend are in agreement. (22,241 vs 22,547).

We thank the reviewer for pointing out this oversight. We have now revised the text accordingly.

Text p. 13: There is still some confusion regarding number of regulatory sites.

In the response to reviewers the number was corrected to 33. The text still reads 31 or 30 (p. 13 and p. 14, p. 19 reads 30?). When you count the number of listed p-sites in the figure it reads 32. Please readdress this issue.

We have now revised the text accordingly.

There are 3 legends to supplementary figure S8. Please correct numbering.

We have now edited the numbering of the legends of the three supplementary figures..

Fig S1B referenced in the text p. 5 reads 0.85-0.99. From the figure it reads 0.68-0.93. Please ensure coherence.

We thank the reviewer for pointing out this oversight. We have now revised the text accordingly.

Fig S1C it is not clear from the figure, legend or text what is z-scored. Ratios? Log2 LFQ?

We have explained in the figure legend that the Log2 LFQ intensity is Z-scored in Fig. S1C.

FigS3B, C is referenced in the text p. 8 to support a “high degree of overlap” between experimental conditions. Please clarify or show support of this statement.

We have linked this statement to the Fig. S3D.

The magnitude of regulation should be made clear by adding numbers to color keys in fig. 3D and 4E.

We have now edited the Figures 3D and 4E, adding numbers to color keys.

A vertical title is missing from fig S5B and as previously mentioned also still for fig S7C.

We have now revised the Figures S5B and S7C.

The reference to figures in the last section of the results does not match. (p.15-16).

We have now revised the manuscript accordingly.

Please make sure that the correct table S4/S5 is referenced on p. 13.

We have now edited the text referring to table S5.

In the discussion Figure 7 is referenced several times. Please check that this is correct.

We have now revised the manuscript accordingly.

Discussion p. 17 please check section 2 for sentence repetitions.

We have now revised the text accordingly

Fig S11 is not referenced and the color key seems off scale.

We have now revised the figure and text accordingly

Text p. 7, one compound is still listed twice.

We have now revised the text accordingly

Text p. 8, what is the calculation behind the statement 65% of the detected proteome?

In Min6 cells we have identified about 9,000 proteins and 5,698 proteins are phosphorylated.

This means that about 65% of the total detected proteome is phosphorylated.

p. 15 please specify in the text what “we also validated the interaction results in a human cell line, HEK293” entails. Which or how many proteins were confirmed?

We thank the reviewer for prompting us to clarify this issue. We have now clarified in the text that in both the experimental system we observed that the S7A mutation impairs the ability to bind almost all the Dnmt3a interactors. This further confirms the role of S7 phosphorylation in the control of the association of the methyltransferase to its partners. In both these two experimental systems we found the H3.1 histone and HDACs (in Hek293 HDAC8 and in Ins1e HDAC2). The small overlap between these two systems is not surprising given the profound differences of the two cell lines HEK293 (Human Embryonic Kidney cells), versus Ins1e1 (Rat Insulinoma cells), and the different levels of Dnmt3a expression observed.

Reviewer #2 (Remarks to the Author):

All of my concerns have been thoroughly addressed in revision.

We are glad that we were able to address the concerns of the reviewer.

Reviewer #3 (Remarks to the Author):

The authors present a very nice revised paper and have adequately addressed my previous concerns, with one exception:

The authors maintain that it is appropriate to refer to MIN6 cells as 'beta-cells' so as to 'emphasize that the work focuses on the GSIS aspect of these cells' (this is a good point I think). However, I disagree with this. MIN6 cells are not beta-cells and to refer to them as such is a bit misleading (one could argue that they are not even the best insulinoma model, but that's beside the point). If the authors wish to emphasize the focus on GSIS, why not refer to "...the drug perturbed phosphoproteome in insulin-secreting cells..."?

We have now revised the text replacing beta cells with “insulin secreting cells”.

References

1. Fuks F, Burgers WA, Godin N, Kasai M, Kouzarides T. Dnmt3a binds deacetylases and is recruited by a sequence-specific repressor to silence transcription. *The EMBO journal* **20**, 2536-2544 (2001).